

# Spider phylogenomics: untangling the Spider Tree of Life

Nicole L. Garrison[1], Juanita Rodriguez[1], Ingi Agnarsson[2],
Jonathan A. Coddington[3], Charles E. Griswold[4], Christopher A. Hamilton[1],
Marshal Hedin[5], Kevin M. Kocot[6], Joel M. Ledford[7] and Jason E. Bond[1]

[1] Department of Biological Sciences and Auburn University Museum of Natural History, Auburn University, Auburn, AL, United States
[2] Department of Biology, University of Vermont, Burlington, VT, United States
[3] Department of Entomology, National Museum of Natural History, Smithsonian Institution, Washingtion, DC, United States
[4] Arachnology, California Academy of Sciences, San Francisco, CA, United States
[5] Department of Biology, San Diego State University, San Diego, CA, United States
[6] Department of Biological Sciences and Alabama Museum of Natural History, University of Alabama—Tuscaloosa, Tuscaloosa, AL, United States
[7] Department of Plant Biology, University of California, Davis, Davis, CA, United States

Corresponding author
Jason E. Bond, jbond@auburn.edu

## ABSTRACT

Spiders (Order Araneae) are massively abundant generalist arthropod predators that are found in nearly every ecosystem on the planet and have persisted for over 380 million years. Spiders have long served as evolutionary models for studying complex mating and web spinning behaviors, key innovation and adaptive radiation hypotheses, and have been inspiration for important theories like sexual selection by female choice. Unfortunately, past major attempts to reconstruct spider phylogeny typically employing the "usual suspect" genes have been unable to produce a well-supported phylogenetic framework for the entire order. To further resolve spider evolutionary relationships we have assembled a transcriptome-based data set comprising 70 ingroup spider taxa. Using maximum likelihood and shortcut coalescence-based approaches, we analyze eight data sets, the largest of which contains 3,398 gene regions and 696,652 amino acid sites forming the largest phylogenomic analysis of spider relationships produced to date. Contrary to long held beliefs that the orb web is the crowning achievement of spider evolution, ancestral state reconstructions of web type support a phylogenetically ancient origin of the orb web, and diversification analyses show that the mostly ground-dwelling, web-less RTA clade diversified faster than orb weavers. Consistent with molecular dating estimates we report herein, this may reflect a major increase in biomass of non-flying insects during the Cretaceous Terrestrial Revolution 125–90 million years ago favoring diversification of spiders that feed on cursorial rather than flying prey. Our results also have major implications for our understanding of spider systematics. Phylogenomic analyses corroborate several well-accepted high level groupings: Opisthothele, Mygalomorphae, Atypoidina, Avicularoidea, Theraphosoidina, Araneomorphae, Entelegynae, Araneoidea, the RTA clade, Dionycha and the Lycosoidea. Alternatively, our results challenge the monophyly of Eresoidea, Orbiculariae, and Deinopoidea. The composition of the major paleocribellate and neocribellate clades, the basal divisions of Araneomorphae, appear to be falsified. Traditional Haplogynae is in need of revision, as our findings appear to support the newly conceived concept of Synspermiata. The sister pairing of filistatids with hypochilids implies that some

![PeerJ]

peculiar features of each family may in fact be synapomorphic for the pair. Leptonetids now are seen as a possible sister group to the Entelegynae, illustrating possible intermediates in the evolution of the more complex entelegyne genitalic condition, spinning organs and respiratory organs.

## INTRODUCTION

Spiders (Order Araneae; Fig. 1) are a prototypical, hyperdiverse arthropod group comprising >45,000 described species (*World Spider Catalog, 2016*) distributed among 3,958 genera and 114 families; by some estimates the group may include >120,000 species (*Agnarsson, Coddington & Kuntner, 2013*). Spiders are abundant, generalist predators that play dominant roles in almost every terrestrial ecosystem. The order represents an ancient group that has continued to diversify taxonomically and ecologically since the Devonian (>380 mya). They are relatively easy to collect and identify, and are one of few large arthropod orders to have a complete online taxonomic catalog with synonymies and associated literature (*World Spider Catalog, 2016*).

In addition to their remarkable ecology, diversity, and abundance, spiders are known for the production of extraordinary biomolecules like venoms and silks as well as their utility as models for behavioral and evolutionary studies (reviewed in *Agnarsson, Coddington & Kuntner, 2013*). Stable and complex venoms have evolved over millions of years to target predators and prey alike. Although few are dangerous to humans, spider venoms hold enormous promise as economically important insecticides and therapeutics (*Saez et al., 2010*; *King & Hardy, 2013*). Moreover, no other animal lineage can claim a more varied and elegant use of silk. A single species may have as many as eight different silk glands, producing a variety of super-strong silks deployed in almost every aspect of a spider's life (*Garb, 2013*): safety lines, dispersal, reproduction (sperm webs, eggsacs, pheromone trails), and prey capture (*Blackledge, Kuntner & Agnarsson, 2011*). Silken prey capture webs, particularly the orb, have long been considered a key characteristic contributing to the ecological and evolutionary success of this group (reviewed in *Bond & Opell, 1998*). Moreover, spider silks are promising biomaterials, already benefiting humans in myriad ways—understanding the phylogenetic basis of such super-materials will facilitate efforts to reproduce their properties in biomimetic materials like artificial nerve constructs, implant coatings, and drug delivery systems (*Blackledge, Kuntner & Agnarsson, 2011*; *Schacht & Scheibel, 2014*).

The consensus on major spider clades has changed relatively little in the last two decades since the summary of *Coddington & Levi (1991)* and *Coddington (2005)*. Under the classical view, Araneae comprises two clades (see Table 1 and Fig. 1 for major taxa discussed throughout; node numbers (Fig. 1) referenced parenthetically hereafter), Mesothelae (Node 2) and Opisthothelae (Node 3). Mesotheles are sisters to all other
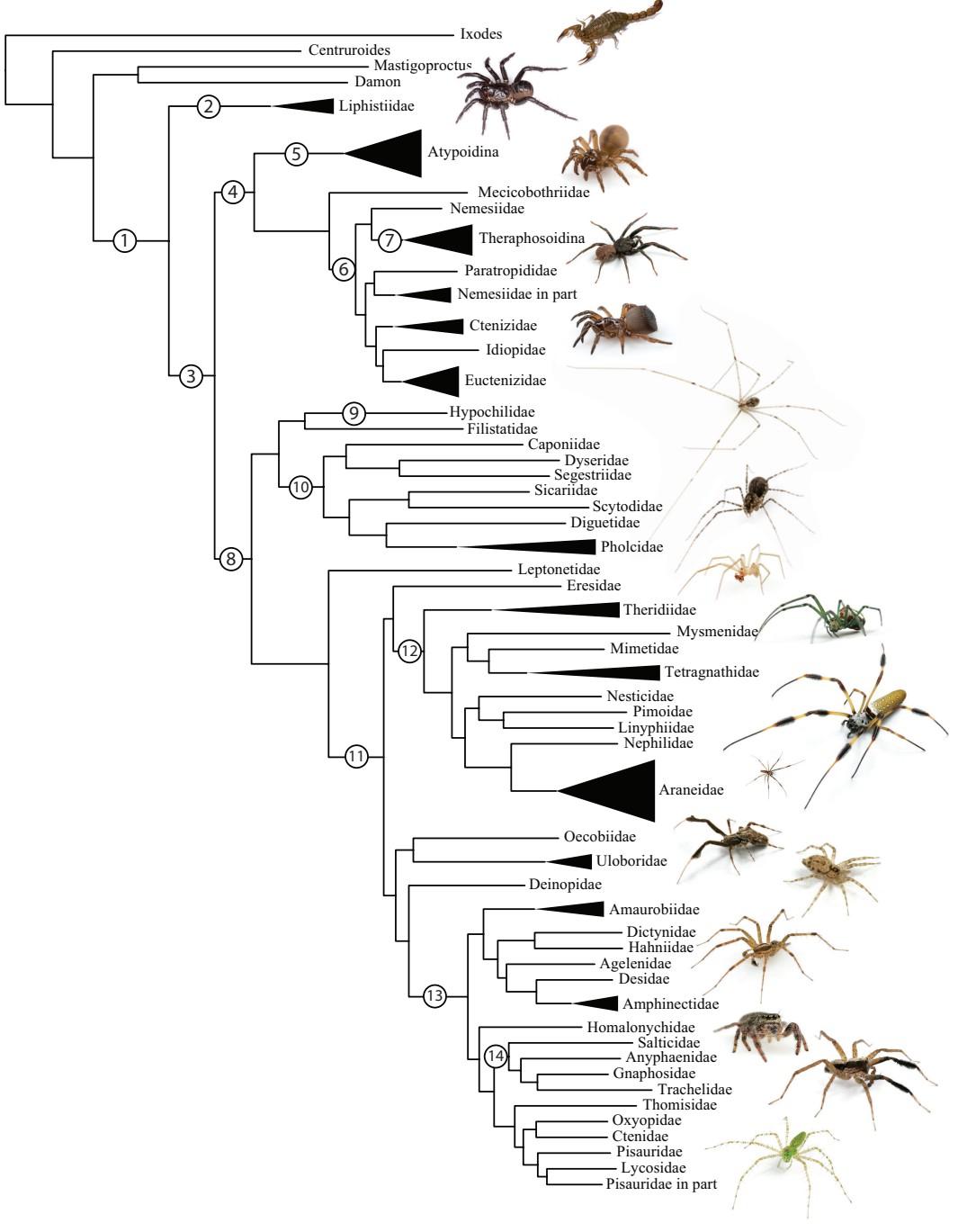

**Figure 1   Summary, preferred tree, of spider relationships based on phylogenomic analyses shown in Fig. 2.** Numbers at nodes correspond to superscripts in Table 1. Images in descending order: Scorpion, Mesothelae, Antrodiaetidae, Paratropididae, Ctenizidae, Pholcidae, Scytodidae, Theridiidae, Tetragnathidae, Nephilidae, Uloboridae, Oecobiidae, Agelenidae, Salticidae, Lycosidae, Oxyopidae.

Peer J

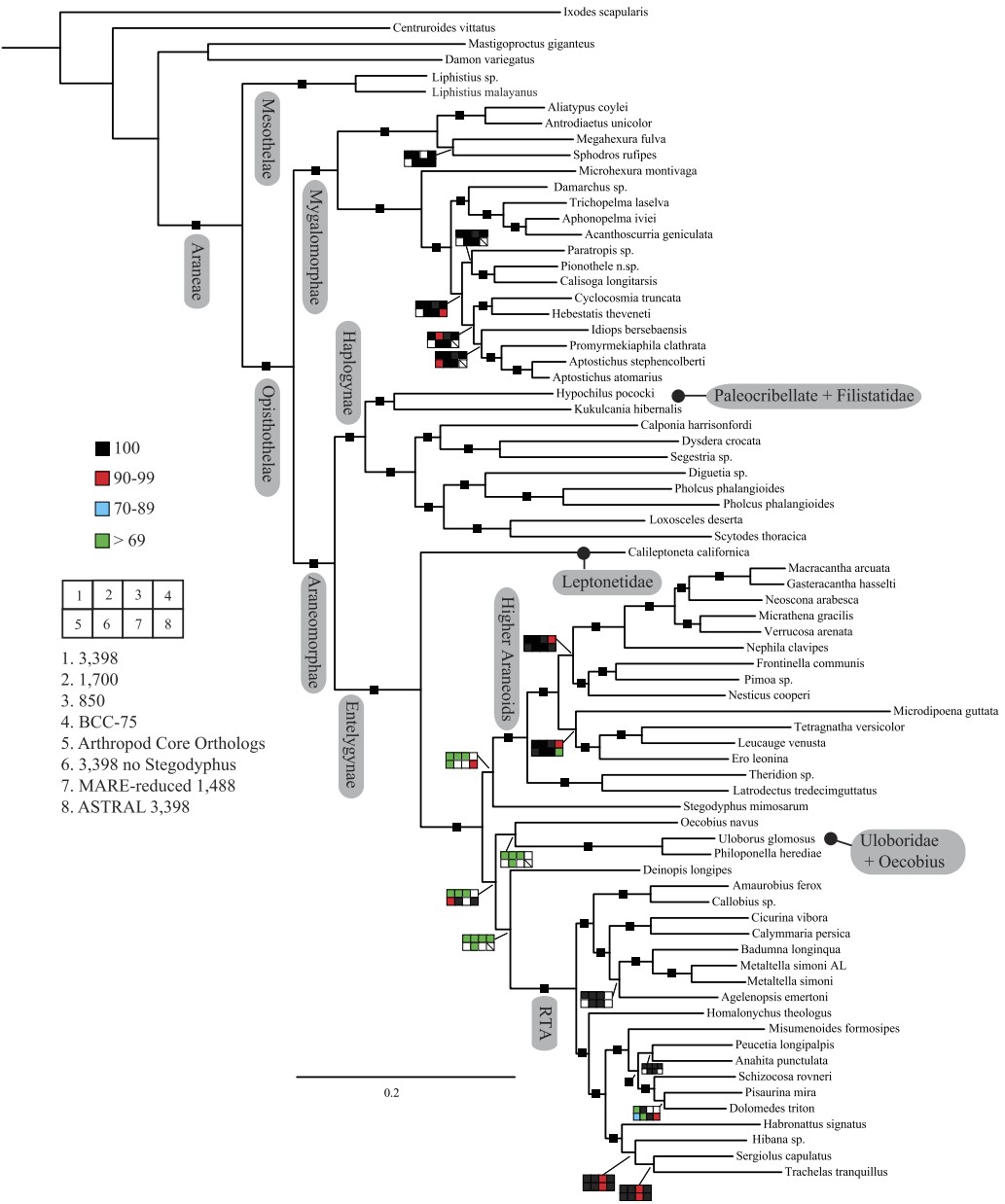

**Figure 2 Summary of phylogenomic analyses (matrices outlined in Table 2) summarized on the phylogenetic hypothesis based on ExaML analysis of dataset 1 (3,398 OGs).** Box plots indicate bootstrap value ranges for each node across matrices 1–7; single solid blocks indicate bootstrap values of 100% in all analyses.

spiders, possessing a plesiomorphic segmented abdomen and mid-ventral (as opposed to terminal) spinnerets. Opisthothelae contains two clades: Mygalomorphae (Node 4) and Araneomorphae (Node 8). Mygalomorphae is less diverse (6% of described Araneae diversity) and retains several plesiomorphic features; e.g., two pairs of book lungs, few and biomechanically 'weak' silks (*Dicko et al., 2008*; *Starrett et al., 2012*). Within Araneomorphae, Hypochilidae (Paleocribellatae; Node 9) is sister to Neocribellatae,

**Table 1  Major spider lineages referenced throughout the text.** Superscripts (column 1) reference node labels in Fig. 1 (summary of family level relationships).

| Lineage | Composition and placement | Description/characteristics |
| --- | --- | --- |
| [1] Araneae | All spiders | Cosmopolitan; cheliceral venom glands, ability to produce silk from abdominal silk glands; male pedipalps modified for sperm transfer |
| [2] Mesothelae | Plesiomorphic sister group to all living spiders | SE Asia; mid ventrally positioned spinnerets; distinct dorsal abdominal tergites, very narrow sternum |
| [3] Opisthothelae | The two major spider lineages | Typical terminal spinneret placement and sternal morphology |
| [4] Mygalomorphae | Trapdoor, baboon and funnel spiders, tarantulas, and their kin | Paraxial chelicerae with venom glands; most lead sedentary lives in burrows; lack anterior median spinnerets; often large and hirsute; two pairs of book lungs |
| [5] Atypoidina | Sister group to remaining mygalomorphs | Most species with vestigial abdominal tergites and unique modifications to male pedipalp |
| [6] Avicularoidea | All remaining mygalomorph taxa | Includes major mygalomorph families, nearly half of which are likely not monophyletic |
| [7] Theraphosoidina | Comprises families Theraphosidae and Barychelidae | Includes the typically large and hirsute tarantulas and baboon spiders |
| [8] Araneomorphae | Over 90% of all spider diversity | Anterior median spinnerets fused to form a cribellum (later lost multiple times) |
| [9] Paleocribellatae | Comprises single family Hypochilidae; hypothesized sister group to all other araneomorphs | Hypochilid synapomorphies, e.g., cheliceral depression; also retain a number of primitive traits including two pairs of booklungs |
| Neocribellatae | Remaining spider lineages | Paracribellum (complimentary spinning field to cribellum); extension of venom gland into prosoma |
| Austrochiloidea | Families Austrochilidae and Gradungulidae; sister group to all other neocribellate lineages | Gondwanan taxa with notched tarsal organs; typically with two pairs of booklungs–posterior pair modified as tracheae in some taxa |
| [10] Haplogynae | Neocribellate lineage with simple genitalia; includes spitting spiders and cellar spiders | Spinnerets lack tartipores; mating with palps inserted simultaneously; in some taxa female genital opening lacks an epigynum; chelicerae fused at base, synspermia, male palpal organ simple |
| [11] Entelegynae | Comprises all remaining spider lineages with complex genitalia | Female genitalia with a flow through system of separate copulatory and fertilization ducts; male palpal organ typically under hydraulic control |
| Palpimanoidea | Comprises a number of enigmatic families | Araneophages with lateral scopulae on anterior legs |
| Eresoidea | Includes 3 families: Eresidae, Hersiliidae, Oecobiidae; sister to remaining entelegynes | Controversial superfamily; oecobiids and hersiliids share a unique attack behavior |
| Orbiculariae | Comprises the Deinopoidea and Araneoidea | Members of this lineage include cribellate and ecribellate orb-web weavers as well as derived araneoids that use adhesive threads to construct sheet and cob-webs |
| Deinopoidea | Includes the cribellate orbicularian families Uloboridae and Deinopidae | Construct cribellate orb web; long considered sister group to adhesive orb web weavers on basis of behavioral web construction data |
| [12] Araneoidea | Spider superfamily that includes adhesive orb web weaving taxa and others | Members of this lineage all use adhesive threads; monophyly supported by a number of spinning and other morphological characteristics |

*(continued on next page)*
**Table 1** (*continued*)

| Lineage | Composition and placement | Description/characteristics |
|---|---|---|
| [13] RTA | Large diverse lineage of spiders that includes wolf, jumping, running, fishing, and crab spiders | Defined primarily by the presence of a projection on the male palp–the retrolateral tibial apophysis (RTA) |
| [14] Dionycha | Subclade of the RTA lineage, comprises about 1/3 of all spider diversity | Defined as a group based on their two clawed condition with flanking tufts of setae for adhesion to smooth surfaces |
| Lycosoidea | Large superfamily comprising 10 families including fishing and wolf spiders | Monophyly of this superfamily is based on a number of morphological features (not universal) including a grate-shaped tapetum, an oval-shaped calamistrum, and male palpal features |

**Table 2**   **Summary of all phylogenomic analyses. Data matrix numbers correspond to Fig. 2, inset.**

| Data set | #OGs | #AAs | % missing | #reps | Log likelihood | Notes |
|---|---|---|---|---|---|---|
| (1) All genes | 3,398 | 696,652 | 38.5% | 225 | −20949310.821967 | ExaML AUTOF |
| (2) 1st reduce | 1,699 | 410,717 | 26.0% | 300 | −14297508.033111 | ExaML AUTOF |
| (3) 2nd reduce | 850 | 230,582 | 19.6% | 300 | −8098715.107390 | ExaML AUTOF |
| (4) BCC | 1,699 | 311,756 | 33.6% | 300 | −10017456.343941 | ExaML AUTOF |
| (5) Arthropod core OG | 549 | 107,307 | 33.0% | 1,000 | −2729523.038858 | ExaML AUTOF bs in RAxML |
| (6) 74 taxa (-Stegodyphus) | 3,398 | 629,566 | 38.8% | 300 | −20569138.970981 | ExaML AUTOF |
| (7) MARE (58 taxa, 55 ingroup) | 1,488 | 351,333 | 19.6% | 295 | −9227466.065087 | ExaML AUTOF |
| (8) ASTRAL | 3,398 | | | 100 | | 100 bootstrap reps per partition |

within which Austrochiloidea is sister to the major clades Haplogynae (Node 10) and Entelegynae (Node 11), each weakly to moderately supported by few morphological features. Haplogynes have simple genitalia under muscular control whereas entelegynes have hydraulically activated, complex genitalia, with externally sclerotized female epigyna. Entelegynes comprise multiple, major, hyperdiverse groups, including the "RTA clade" (RTA = retrolateral tibial apophysis, Node 13), its subclade Dionycha (e.g., jumping spiders; *Ramírez, 2014*, Node 14), and the Orbiculariae—the cribellate and ecribellate orb weavers and relatives (see *Hormiga & Griswold, 2014*).

Beginning with early higher-level molecular phylogenetic studies, it gradually became clear that major "stalwart" and presumably well-supported spider groups like the Neocribellatae, Haplogynae, Palpimanoidea, Orbiculariae, Lycosoidea, and others (generally only known to arachnologists) were questionable. Subsequent studies focusing on mygalomorph (*Hedin & Bond, 2006*; *Bond et al., 2012*) and araneomorph (*Blackledge et al., 2009*; *Dimitrov et al., 2012*) relationships continued to challenge the consensus view based largely on morphological data, finding polyphyletic families and ambivalent support for major clades, which were sometimes "rescued" by adding non-molecular data; molecular signal persistently contradicted past verities. In *Agnarsson, Coddington & Kuntner (2013)*, a meta-analysis of available molecular data failed to recover several major groups such as Araneomorphae, Haplogynae, Orbiculariae, Lycosoidea, and others (Table 1). Although these authors criticized the available molecular data as insufficient, their results actually presaged current spider phylogenomic inferences (*Bond et al., 2014*).

Incongruence between the traditional spider classification scheme and (non-phylogenomic) molecular systematics likely has one primary cause: too few data. Non-molecular datasets to date have been restricted to a relatively small set of morphological and/or behavioral characters whereas molecular analyses addressing deep spider relationships have largely employed relatively few, rapidly evolving loci (e.g., 28S and 18S rRNA genes, Histone 3, and a number of mitochondrial DNA markers).

The first analyses of spider relationships using genome-scale data, scored for 40 taxa by *Bond et al. (2014)* and for 14 taxa by *Fernández, Hormiga & Giribet (2014)*, considerably refined understanding of spider phylogeny, the former explicitly calling into question long held notions regarding the tempo and mode of spider evolution. Using transcriptome-derived data, *Bond et al. (2014)* recovered the monophyly of some major groups (araneomorphs and mygalomorphs) but reshuffled several araneomorph lineages (haplogynes, paleocribellates, orbicularians, araneoids (Node 12) and the RTA clade). Notably, *Bond et al. (2014)* and *Fernández, Hormiga & Giribet (2014)* rejected Orbiculariae, which included both cribellate (Deinopoidea) and ecribellate orb weavers (Araneoidea). Instead they suggested either that the orb web arose multiple times, or, more parsimoniously, that it arose once and predated the major diversification of spiders. Despite significant advances in understanding of spider phylogeny, only a small percentage of spider families were sampled and monophyly of individual families could not be tested in previous phylogenomic studies. Denser taxon sampling is needed to warrant changes in higher classification and to more definitively address major questions about spider evolution.

Herein, we apply a spider-specific core ortholog approach with significantly increased taxon and gene sampling to produce a more complete and taxon specific set of alignments for phylogenetic reconstruction and assessment of spider evolutionary pattern and process. Existing genome-derived protein predictions and transcriptome sequences from a representative group of spiders and arachnid outgroups were used to create a custom core ortholog set specific to spiders. Taxon sampling was performed to broadly sample Araneae with an emphasis on lineages whose phylogenetic placement is uncertain and included previously sequenced transcriptomes, gene models from completely sequenced genomes, and novel transcriptome sequences generated by our research team. This resulted in a data set comprising 70 spider taxa plus five additional arachnid taxa as outgroups. We test long-held notions that the orb web, in conjunction with ecribellate adhesive threads, facilitated diversification among araneoids and present the most completely sampled phylogenomic data set for spiders to date using an extensive dataset of nearly 3,400 putative genes (~700 K amino acids). Further, we test the hypothesis of a non-monophyletic Orbiculariae, assess diversification rate shifts across the spider phylogeny, and provide phylogenomic hypotheses for historically difficult to place spider families. Our results clearly demonstrate that our understanding of spider phylogeny and evolution requires major reconsideration and that several long-held and contemporary morphologically-derived hypotheses are likely destined for falsification.

## MATERIALS & METHODS

### Sampling, extraction, assembly

Spider sequence data representing all major lineages were collected from previously published transcriptomic and genomic resources ($N = 53$) and supplemented with newly sequenced transcriptomes ($N = 21$) to form the target taxon set for the current study. Existing sequence data were acquired via the NCBI SRA database (http://www.ncbi.nlm.nih.gov/sra). Raw transcriptome sequences were downloaded, converted to fastq file format, and assembled using Trinity (*Grabherr et al., 2011*). Genomic data sets in the form of predicted proteins were downloaded directly from the literature (*Sanggaard et al., 2014*) for downstream use in our pipeline. Newly sequenced spiders were collected from a variety of sources, extracted using the TRIzol total RNA extraction method, purified with the RNeasy mini kit (Qiagen) and sequenced in-house at the Auburn University Core Genetics and Sequencing Laboratory using an Illumina Hi-Seq 2500. This produced 100bp paired end reads for each newly sequenced spider transcriptome, which were then assembled using Trinity. Proteins were predicted from each transcriptome using the program TransDecoder (*Haas et al., 2013*).

### Core ortholog approach and data processing

We employed a core ortholog approach for putative ortholog selection and implicitly compared the effect of using a common arthropod core ortholog set and one compiled for spiders; the arthropod core ortholog set was deployed as described in *Bond et al. (2014)*. To generate the spider core ortholog set, we used an all-versus-all BLASTP method (*Altschul et al., 1990*) to compare the transcripts of the amblypygid *Damon variegatus*, and the spiders *Acanthoscurria geniculata*, *Dolomedes triton*, *Ero leonina*, *Hypochilus pococki*, *Leucauge venusta*, *Liphistius malayanus*, *Megahexura fulva*, *Neoscona arabesca*, *Stegodyphus mimosarum*, and *Uloborus sp.*. *Acanthoscurria geniculata* and *Stegodyphus mimosarum* were represented by predicted transcripts from completely sequenced genomes while the other taxa were represented by our new Illumina transcriptomes. An e-value cut-off of 10–5 was used. Next, based on the BLASTP results, Markov clustering was conducted using OrthoMCL 2.0 (*Li, Stoeckert & Roos, 2003*) with an inflation parameter of 2.1.

The resulting putatively orthologous groups (OGs) were processed with a modified version of the bioinformatics pipeline employed by *Kocot et al. (2011)*. First, sequences shorter than 100 amino acids in length were discarded. Next, each candidate OG was aligned with MAFFT (*Katoh, 2005*) using the automatic alignment strategy with a maxiterate value of 1,000. To screen OGs for evidence of paralogy, an "approximately maximum likelihood tree" was inferred for each remaining alignment using FastTree 2 (*Price, Dehal & Arkin, 2010*). Briefly, this program constructs an initial neighbor-joining tree and improves it using minimum evolution with nearest neighbor interchange (NNI) subtree rearrangement. FastTree subsequently uses minimum evolution with subtree pruning regrafting (SPR) and maximum likelihood using NNI to further improve the tree. We used the "slow" and "gamma" options; "slow" specifies a more exhaustive NNI search, while "gamma" reports the likelihood under a discrete gamma approximation with 20 categories, after the final round of optimizing branch lengths. PhyloTreePruner (*Kocot, Citarella & Halanych, 2013*)

was then employed as a tree-based approach to screen each candidate OG for evidence of paralogy. First, nodes with support values below 0.95 were collapsed into polytomies. Next, the maximally inclusive subtree was selected where all taxa were represented by no more than one sequence or, in cases where more than one sequence was present for any taxon, all sequences from that taxon formed a monophyletic group or were part of the same polytomy. Putative paralogs (sequences falling outside of this maximally inclusive subtree) were then deleted from the input alignment. In cases where multiple sequences from the same taxon formed a clade or were part of the same polytomy, all sequences but the longest were deleted. Lastly, in order to eliminate orthology groups with poor taxon sampling, all groups sampled for fewer than 7 of the 11 taxa and all groups not sampled for *Megahexura fulva* (taxon with greatest number of identified OGs) were discarded. The remaining alignments were used to build profile hidden Markov models (pHMMs) for HaMStR with hmmbuild and hmmcalibrate from the HMMER package (*Eddy, 2011*).

For orthology inference, we employed HaMStR v13.2.3 (*Ebersberger, Strauss & Von Haeseler, 2009*), which infers orthology based on predefined sets of orthologs. Translated transcripts for all taxa were searched against the new set of 4,934 spider-specific pHMMs (available for download from the Dryad Data Repository) and an arthropod core ortholog set previously employed in *Bond et al. (2014)*. In the spider core ortholog analysis, the genome-derived *Acanthoscurria geniculata* OGs were used as the reference protein set for reciprocal best hit scoring. *Daphnia pulex* was used as the reference species for putative ortholog detection in the arthropod core ortholog analysis. Orthologs sharing a core identification number were pooled together for all taxa and processed using a modified version of the pipeline used to generate the custom spider ortholog set. In both analyses, sequences shorter than 75 amino acids were deleted first. OGs sampled for fewer than 10 taxa were then discarded. Redundant identical sequences were removed with the perl script uniqhaplo.pl (available at http://raven.iab.alaska.edu/~ntakebay/) leaving only unique sequences for each taxon. Next, in cases where one of the first or last 20 characters of an amino acid sequence was an X (corresponding to a codon with an ambiguity, gap, or missing data), all characters between the X and that end of the sequence were deleted and treated as missing data. Each OG was then aligned with MAFFT (mafft—auto—localpair—maxiterate 1,000; *Katoh (2005)*). Alignments were then trimmed with ALISCORE (*Misof & Misof, 2009*) and ALICUT (*Kück, 2009*) to remove ambiguously aligned regions. Next, a consensus sequence was inferred for each alignment using the EMBOSS program infoalign (*Rice, Longden & Bleasby, 2000*). For each sequence in each single-gene amino acid alignment, the percentage of positions of that sequence that differed from the consensus of the alignment were calculated using infoalign's "change" calculation. Any sequence with a "change" value greater than 75 was deleted. Subsequently, a custom script was used to delete any mistranslated sequence regions of 20 or fewer amino acids in length surrounded by ten or more gaps on either side. This step was important, as sequence ends were occasionally mistranslated or misaligned. Alignment columns with fewer than four non-gap characters were subsequently deleted. At this point, alignments shorter than 75 amino acids in length were discarded. Lastly, we deleted sequences that did not overlap with all other sequences in the alignment by at least 20 amino acids, starting with the

shortest sequence not meeting this criterion. This step was necessary for downstream single-gene phylogenetic tree reconstruction. As a final filtering step, OGs sampled for fewer than 10 taxa were discarded.

In some cases, a taxon was represented in an OG by two or more sequences (splice variants, lineage-specific gene duplications (= inparalogs), overlooked paralogs, or exogenous contamination). In order to select the best sequence for each taxon and exclude any overlooked paralogs or exogenous contamination, we built trees in FastTree 2 (*Price, Dehal & Arkin, 2010*) and used PhyloTreePruner to select the best sequence for each taxon as described above. Remaining OGs were then concatenated using FASconCAT (*Kück & Meusemann, 2010*). The OGs selected by our bioinformatic pipeline were further screened in seven different ways (subsets listed in Table 2). OGs were first sorted based on amount of missing data; the half with the lowest levels was pulled out as matrix 2 (1,699 genes). From matrix 2, a smaller subset of OGs optimized for gene occupancy was extracted, resulting in matrix 3 (850 genes). The full supermatrix (matrix 1) was also optimized using the programs MARE (*Meyer, Meusemann & Misof, 2011*) and BaCoCa (Base Composition Calculator; *Kück & Struck, 2014*). MARE assesses the supermatrix by partition, providing a measure of tree-likeness for each gene and optimizes the supermatrix for information content. The full supermatrix was optimized with an alpha value of 5, to produce matrix 7 (1,488 genes, 58 taxa). From the MARE-reduced matrix, genes having no missing partitions for any of the remaining taxa ($n = 50$) were extracted to form a starting matrix for the BEAST analyses (details below). Matrix assessment was also conducted using BaCoCa, which provides a number of descriptive supermatrix statistics for evaluating bias in amino acid composition and patterns in missing data. This program was used to assess patterns of non-random clusters of sequences in the data, which could potentially mislead phylogenetic analyses. Matrix 4 represents a 50% reduction of the full supermatrix using BaCoCa derived values for phylogenetically informative sites as a guide; essentially reducing missing data from absent partitions and gaps. This matrix is similar, but not identical to matrix 2. Matrix 5 resulted from application of arthropod core OGs from *Bond et al. (2014)* to the extended taxon set. Matrix 6 represents the full spider core OG matrix (matrix 1) with *Stegodyphus* pruned from the tree. OGs for each matrix were concatenated using FASconCAT (*Kück & Meusemann, 2010*).

## Phylogenetics

Table 2 summarizes run parameters of the seven individual maximum likelihood analyses conducted for each of the supermatrices. We selected the optimal tree for each supermatrix using the computer program ExaML ver. 3.0.1 (*Kozlov, Aberer & Stamatakis, 2015*). Models of amino acid substitution were selected using the AUTOF command in ExaML. Bootstrap data sets and starting parsimony trees for each matrix were generated using RAxML (*Stamatakis, 2014*) and each individually analyzed in ExaML. We generated 225–300 replicates for each matrix which were then used to construct a majority-rule bootstrap consensus tree; a custom python script was used to automate the process and write a bash script to execute the analyses on a high performance computing (HPC) cluster. The arthropod core OG bootstrap analysis was conducted using RAxML. All
analyses were conducted on the Auburn University CASIC HPC and Atrax (Bond Lab, Auburn University).

A coalescent-based method as implemented in ASTRAL (Accurate Species TRee ALgorithm; *Mirarab et al., 2014*) was used to infer a species tree from a series of unrooted gene trees. The ASTRAL approach is thought to be more robust to incomplete lineage sorting, or deep coalescence, than maximum likelihood analysis of concatenated matrices and works quickly on genome-scale datasets (*Mirarab et al., 2014*). We first constructed individual gene trees for all partitions contained within matrix 1. Gene trees were generated using ML based on 100 RAxML random addition sequence replicates followed by 100 bootstrap replicates (Table 2). Subsequent species tree estimation was inferred using ASTRAL v4.7.6, from all individual unrooted gene trees (and bootstrap replicates), under the multi-species coalescent model.

A chronogram was inferred in a Bayesian framework under an uncorrelated lognormal relaxed clock model (*Drummond et al., 2006*; *Drummond, 2007*) using Beast v1.8.1 (*Drummond et al., 2012*). For this analysis we used 43 partitions of a matrix which included complete partitions for all taxa derived from the MARE-optimized matrix 7. The model of protein evolution for each partition was determined using the perl script ProteinModelSelection.pl in RAxML. BEAST analyses were run separately for each partition using eight calibration points based on fossil data. The most recent common ancestor (MRCA) of Mesothelae + all remaining spiders was given a lognormal prior of (mean in real space) 349 Ma (SD = 0.1) based on the Mesothelae fossil *Palaeothele montceauensis* (*Selden, 1996*). The MRCA of extant araneomorphs was given a lognormal prior of (mean in real space) 267 Ma (SD = 0.2) based on the fossil *Triassaraneus andersonorum* (*Selden et al., 1999*). The MRCA of extant mygalomorphs was given a lognormal prior of (mean in real space) 278 Ma (SD = 0.1) based on the fossil *Rosamygale grauvogeli* (*Selden & Gall, 1992*). The MRCA of Haplogynae + Hypochilidae was given a lognormal prior of (mean in real space) 278 Ma (SD = 0.1) based on the fossil *Eoplectreurys gertschi* (*Selden & Penney, 2010*). The MRCA of Deinopoidea (cribellate orb-weavers) was given a lognormal prior of (mean in real space) 195 Ma (SD = 0.3) based on the fossil *Mongolarachne jurassica* (*Selden, Shih & Ren, 2013*). The MRCA of ecribellate orb-weavers was given a lognormal prior of (mean in real space) 168 Ma (SD = 0.4) based on the fossil *Mesozygiella dunlopi* (*Penney & Ortu, 2006*). The MRCA of Nemesiidae, excluding *Damarchus*, was given a lognormal prior of (mean in real space) 168 Ma (SD = 0.4) based on the nemesiid fossil *Cretamygale chasei* (*Selden, 2002*). Finally, the MRCA of Antrodiaetidae was given a lognormal prior of (mean in real space) 168 Ma (SD = 0.4) based on the fossil *Cretacattyma raveni* (*Eskov & Zonstein, 1990*). Two or more independent Markov Chain Monte Carlo (MCMC) searches were performed until a parameter effective sample size (ESS) >200 was achieved. ESS values were examined in Tracer v1.5. Independent runs for each partition were assembled with LogCombiner v1.7.5 and 10% percent of generations were discarded as burn-in. Tree files for each partition where then uniformly sampled to obtain 10,000 trees. A total of 430,000 trees (10,000 trees from each partition) were assembled with LogCombiner v1.7.5 and a consensus tree was produced using TreeAnnotator v1.8.1. A chronogram containing all taxa was generated using a penalized likelihood method in r8s v1.8 (*Sanderson, 2002*). The

95% highest posterior density dates obtained for the BEAST analysis were incorporated as constraints for node ages of the eight fossil calibrated nodes. The analysis was performed using the TN algorithm, cross validation of branch-length variation and rate variation modeled as a gamma distribution with an alpha shape parameter.

To detect diversification rate shifts, we performed a Bayesian analysis of diversification in BAMM (Bayesian Analysis of Macroevolutionary Mixtures; *Rabosky et al., 2014*). For this analysis we used the chronogram obtained by the r8s analysis in order to maximize taxon sampling. To account for non-random missing speciation events, we quantified the percentage of taxa sampled per family (*World Spider Catalog, 2015*) and incorporated these into the analysis. We also accounted for missing families sampled. The MCMC chain was run for 100,000,000 generations, with sampling every 10,000 generations. Convergence diagnostics were examined using coda (*Plummer et al., 2006*) in R. Ten percent of the runs were discarded as burn-in. The 95% credible set of shift configurations was plotted in the R package BAMMtools (*Rabosky et al., 2014*).

Character state reconstructions of web type following *Blackledge et al. (2009)* were performed using a maximum likelihood approach. The ML approach was implemented using the rayDISC command in the package corHMM (*Beaulieu, O'Meara & Donoghue, 2013*) in R (*Ihaka & Gentleman, 1996*). This method allows for multistate characters, unresolved nodes, and ambiguities (polymorphic taxa or missing data). Three models of character evolution were evaluated under the ML method: equal rates (ER), symmetrical (SYM) and all rates different (ARD). A likelihood-ratio test was performed to select among these varying models of character evolution.

# RESULTS

## Summary of genomic data

Twenty-one novel spider transcriptomes were sequenced, with an average of 72,487 assembled contigs (contiguous sequences) ranging from 6,816 (*Diguetia sp.*) to 191,839 (*Segestria sp.*); specimen data and transcriptome statistics for each sample are summarized in Tables S1 and S2 respectively. Median contig length for the novel transcriptomes was 612 bp. The complete taxon set, including spider and outgroup transcriptomes from the SRA database, had an average contig number of 53,740 and a range of 5,158 (*Paratropis sp.*) to 202,311 (*Amaurobius ferox*) with a median contig length of 655. The newly constructed spider-specific core ortholog group (OG) set contained 4,934 OGs, more than three times the number of arthropod core orthologs used in prior spider analyses (*Bond et al., 2014*) and represents a significant step forward in generating a pool of reasonably well-vetted orthologs for spider phylogenomic analyses. The arthropod and spider core orthology sets had 749 groups in common; 4,185 OGs in the spider core were novel. Of the spider-core groups, 4,249 (86%) were present in the sequenced genome of our HaMSTR reference taxon of choice *Acanthoscurria geniculata* (*Sanggaard et al., 2014*) and were retained for use in downstream ortholog detection. The number of TransDecoder predicted proteins and ortholog detection success for each taxon is summarized in Table S2. Annotations for the arthropod set can be found in *Bond et al. (2014)*; Table S3 summarizes gene annotations
for the spider core ortholog set generated for this study. Our new HaMStR spider core ortholog set and *Acanthoscurria geniculata* BLAST database file can be downloaded from the Dryad Data Repository at doi:10.5061/dryad.6p072.

## Phylogenetic analyses

Seven super matrices were generated for downstream non time-calibrated analyses (Fig. 2), one drawn from the arthropod core set and six using the spider core set. Data set sizes, summarized in Table 2, ranged from a maximum of 3,398 OGs with a higher percentage of missing cells (38.5%), 850 OGs with 19.6% missing, to 549 OGs (arthropod core set) with 33% missing data. Two matrices were generated using automated filtering approaches implemented by BaCoCa (*Kück & Struck, 2014*) and MARE (*Meyer, Meusemann & Misof, 2011*). In BaCoCa we sorted partitions using number of informative sites, capturing the top half (1,700 OGs) of the matrix containing the most informative sites. RCFV values generated by BaCoCa were <0.05 for all taxa in all partitions for each of the matrices, indicating homogeneity in base composition. Additionally, there was no perceptible taxonomic bias observed in shared missing data (Figs. S1–S6). The MARE optimized matrix comprised 58 taxa and 1,488 genes with 19.6% missing data. For graphical representations of gene occupancy for each matrix, see Figs. S7–S12. Blast2GO (*Conesa et al., 2005*) gene ontology distributions of molecular function for OGs recovered from both the spider and arthropod ortholog sets (Figs. S13 and S14) can be found in the Supplemental Information.

Our phylogenetic analyses (see Table 2 and 'Discussion'), the results of which are summarized in Fig. 2, consistently recover many well-supported monophyletic groups: Araneae, Mygalomorphae, Araneomorphae, Synspermiata (i.e., Haplogynae excluding Filistatidae and Leptonetidae), Entelegynae, the RTA clade, Dionycha, and Lycosoidea. Within Mygalomorphae, Atypoidina and Avicularioidea are monophyletic; Nemesiidae is polyphyletic. Filistatidae (*Kukulcania*) emerges as the sister group to *Hypochilus*. Interestingly, Leptonetidae emerges as the sister group to Entelegynae. Eresidae is sister to Araneoidea, similar to findings of *Miller et al. (2010)*. Deinopoidea is polyphyletic. Oecobiidae is sister to Uloboridae, which are together sister to Deinopidae plus the RTA clade. Homalonychidae and by implication the entire Zodarioidea (*Miller et al., 2010*), is sister to Dionycha plus Lycosoidea. Hahniidae, represented by the cryphoecine *Calymmaria*, is sister to Dictynidae. Thomisidae belongs in Lycosoidea as proposed by *Homann (1971)* and *Polotow, Carmichael & Griswold (2015)* (see also *Ramírez, 2014*).

Coalescent-based species-tree analysis in ASTRAL employed unrooted gene trees based on the 3,398 gene matrix as input and inferred a well-supported tree (most nodes >95% bs; Fig. 3). With few exceptions the topology recovered using this approach was congruent with the likelihood-based supermatrix analysis. Conflicting nodes, some corresponding to key araneomorph lineages, which were moderately to weakly supported in concatenated analyses, are summarized in Fig. 2.

A chronogram based on 43 partitions with no missing data (matrix 7, see Table 2) is shown in Fig. 4. MRCA divergence time estimates are summarized in Table 3: Mesothelae—Opisthothelae at 340 Ma (95% CI[287–398]); Mygalomorphae—Araneomorphae at 308 Ma (95% CI[258–365]); Synspermiata + Hypochilidae—Entelegynae at 276 Ma
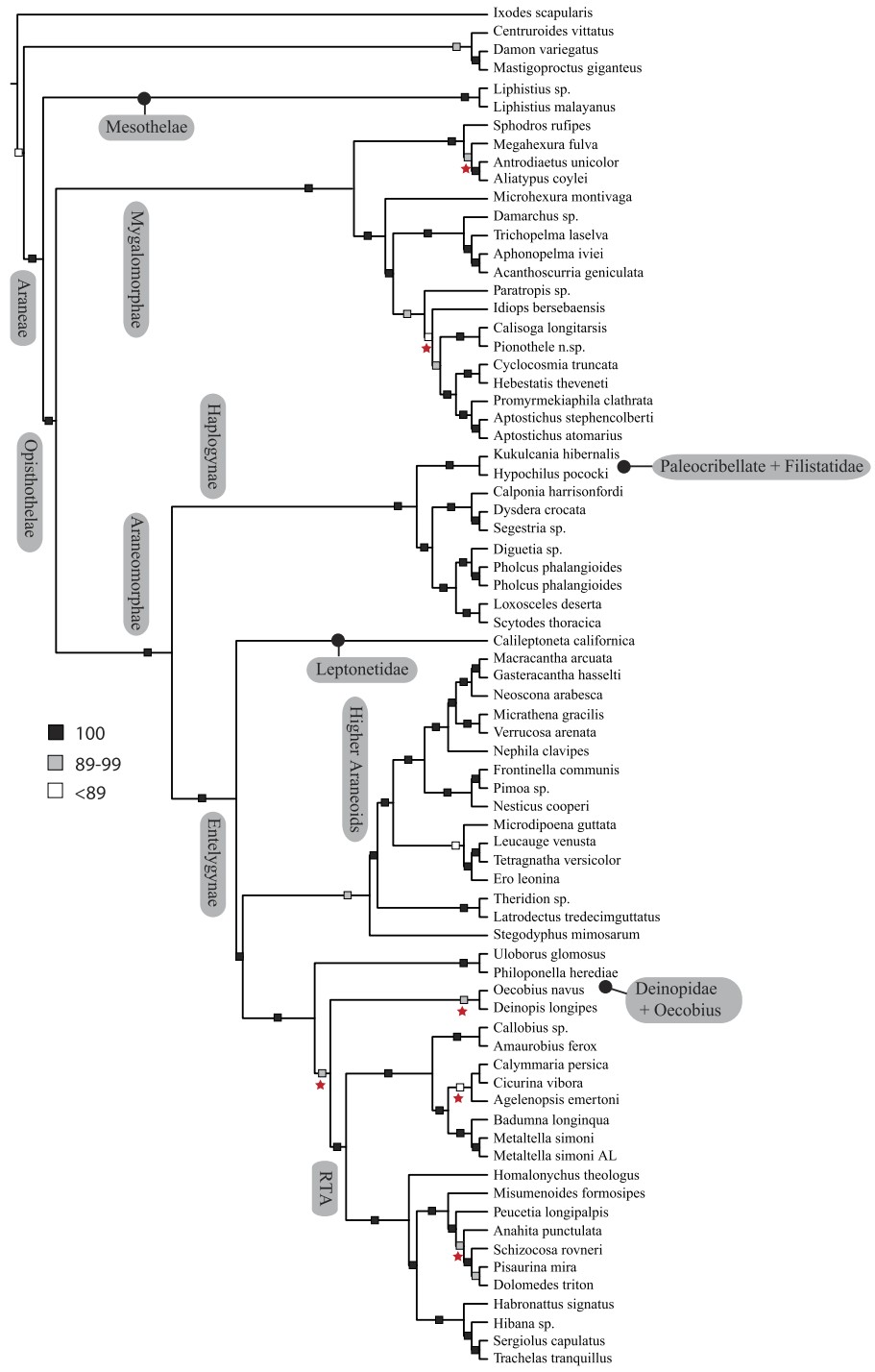

**Figure 3  ASTRAL gene tree analysis of spider relationships based on 3,398 genes.** Relative support value ranges reported at each node (inset legend); red stars indicate branches not congruent with tree shown in Figs. 1 and 2.

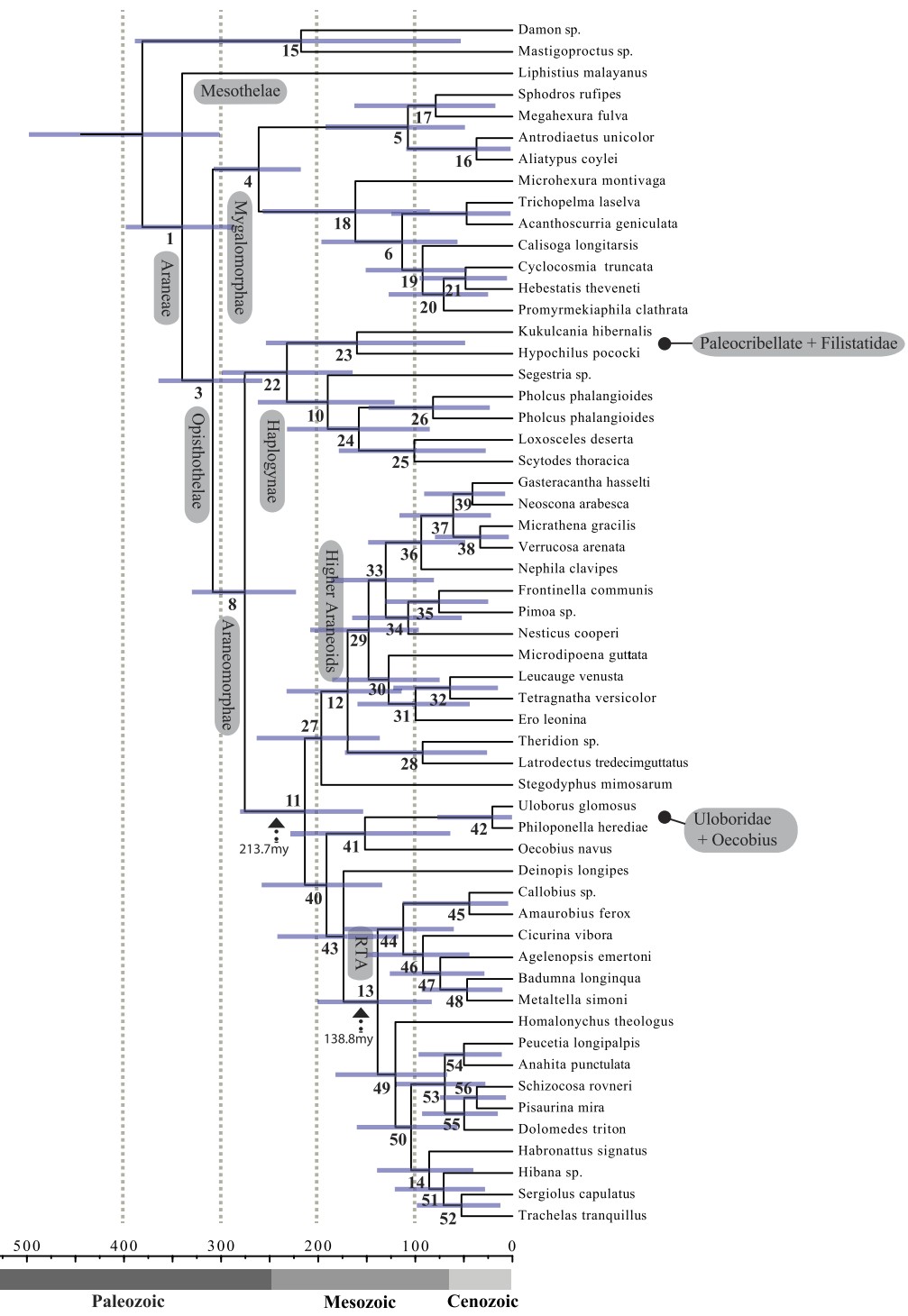

**Figure 4  Chronogram resulting from two Bayesian MCMC runs performed in BEAST showing estimated divergence time for major spider lineages.** Time scale on *x* axis; node point estimates and 95% confidence intervals (blue bars) are reported in Table 2. Node numbers correspond to numbering scheme used in Tables 1 and 2.

**Table 3 Posterior probabilities (PP), ages (Ma), and 95% confidence intervals (CI) for the highest posterior density (HPD) recovered by the BEAST analysis.** Node numbers correspond to Fig. 5. Node numbers in bold correspond to numbers in Fig. 1 and Table 1.

| Node | Age | HPD 95% CI | Taxonomic group |
|------|-----|-----------|-----------------|
| **1** | 340 | 287–398 | Araneae |
| **3** | 309 | 258–365 | Opisthele |
| **4** | 261 | 218–307 | Mygalomorphae |
| **5** | 108 | 49–192 | Atypoidina |
| **6** | 114 | 57–197 | Avicularoidea |
| **7** | 47 | 2–125 | Theraphosoidina |
| **8** | 276 | 223–330 | Opisthelae |
| **10** | 190 | 121–262 | Haplogynae |
| **11** | 214 | 154–280 | Entelegynae |
| **12** | 170 | 114–233 | Araneoidea |
| **13** | 139 | 83–201 | RTA |
| **14** | 86 | 40–139 | Dionycha |
| 15 | 218 | 53–389 | |
| 16 | 37 | 2–109 | |
| 17 | 79 | 18–163 | |
| 18 | 162 | 85–257 | |
| 19 | 93 | 47–151 | |
| 20 | 71 | 25–127 | |
| 21 | 48 | 35–217 | Ctenizidae |
| 22 | 232 | 165–299 | |
| 23 | 160 | 49–254 | |
| 24 | 158 | 85–232 | |
| 25 | 101 | 28–179 | |
| 26 | 81 | 23–148 | Pholcidae |
| 27 | 197 | 137–263 | |
| 28 | 92 | 26–172 | Theridiidae |
| 29 | 148 | 96–208 | |
| 30 | 127 | 75–186 | |
| 31 | 100 | 44–160 | |
| 32 | 64 | 15–123 | Tetragnathidae |
| 33 | 130 | 81–186 | |
| 34 | 107 | 52–165 | |
| 35 | 76 | 25–131 | |
| 36 | 94 | 49–149 | |
| 37 | 61 | 22–116 | Araneidae |
| 38 | 33 | 29–312 | |
| 39 | 41 | 33–420 | |
| 40 | 191 | 134–258 | |
| 41 | 152 | 64–228 | |

**Table 3** (*continued*)

| Node | Age | HPD 95% CI | Taxonomic group |
|------|-----|------------|-----------------|
| 42 | 21 | 28–126 | Uloboridae |
| 43 | 174 | 117–242 | |
| 44 | 112 | 60–174 | |
| 45 | 44 | 4–113 | |
| 46 | 92 | 44–149 | |
| 47 | 74 | 29–126 | |
| 48 | 47 | 34–243 | |
| 49 | 120 | 68–182 | |
| 50 | 104 | 57–160 | |
| 51 | 71 | 28–121 | |
| 52 | 52 | 36–130 | |
| 53 | 70 | 28–120 | Lycosoidea |
| 54 | 50 | 35–735 | |
| 55 | 49 | 15–93 | |
| 56 | 37 | 27–211 | |

(95% CI[223–330]); RTA + Deinopoidea—*Stegodyphus* + Araneoidea at 214 Ma (95% CI[154–280]); RTA—Dionycha at 138.8 Ma (Fig. 4).

Diversification rate shift analysis estimated three instances of significant diversification shifts within spiders (95% credibility). The highest rate shift is within the RTA + Dionycha + Lycosoidea (Fig. 5) followed by Avicularioidea and within Araneoidea ($f = 0.23$; 0.21; Fig. 5).

Maximum likelihood ancestral state reconstruction of web type (Fig. 6) shows that the spider common ancestor likely foraged from a subterranean burrow, sometimes sealed by a trapdoor. The ancestral condition for araneomorphs may have been a stereotypical aerial sheet. Entelegynae ancestors probably spun orbs, which were subsequently lost at least three times. RTA taxa largely abandoned webs to become hunting spiders. Precise location of these character state shifts depends upon sufficient sampling; denser sampling reduces the number of unobserved evolutionary events. While this analysis contains only 47 of 114 spider families, the sequence and overall mapping to the spider backbone phylogeny is strongly supported.

## DISCUSSION

Our phylogenomic analyses represent the largest assessment of spider phylogeny to date using genomic data, both in terms of taxa and number of orthologs sampled. Our results are largely congruent with earlier work (*Bond et al., 2014*): we recover all of the major backbone lineages (Mygalomorphae, Araneomorphae, RTA, etc.), but reiterate that our understanding of spider evolutionary pattern and process needs thorough reconsideration. This expanded study reinforces the ancient origin of the orb web hypothesis (*Bond et al., 2014*) and shows that rates of spider species diversification appear to be associated with web change or loss—or with modification of the male palp rather than the origin of

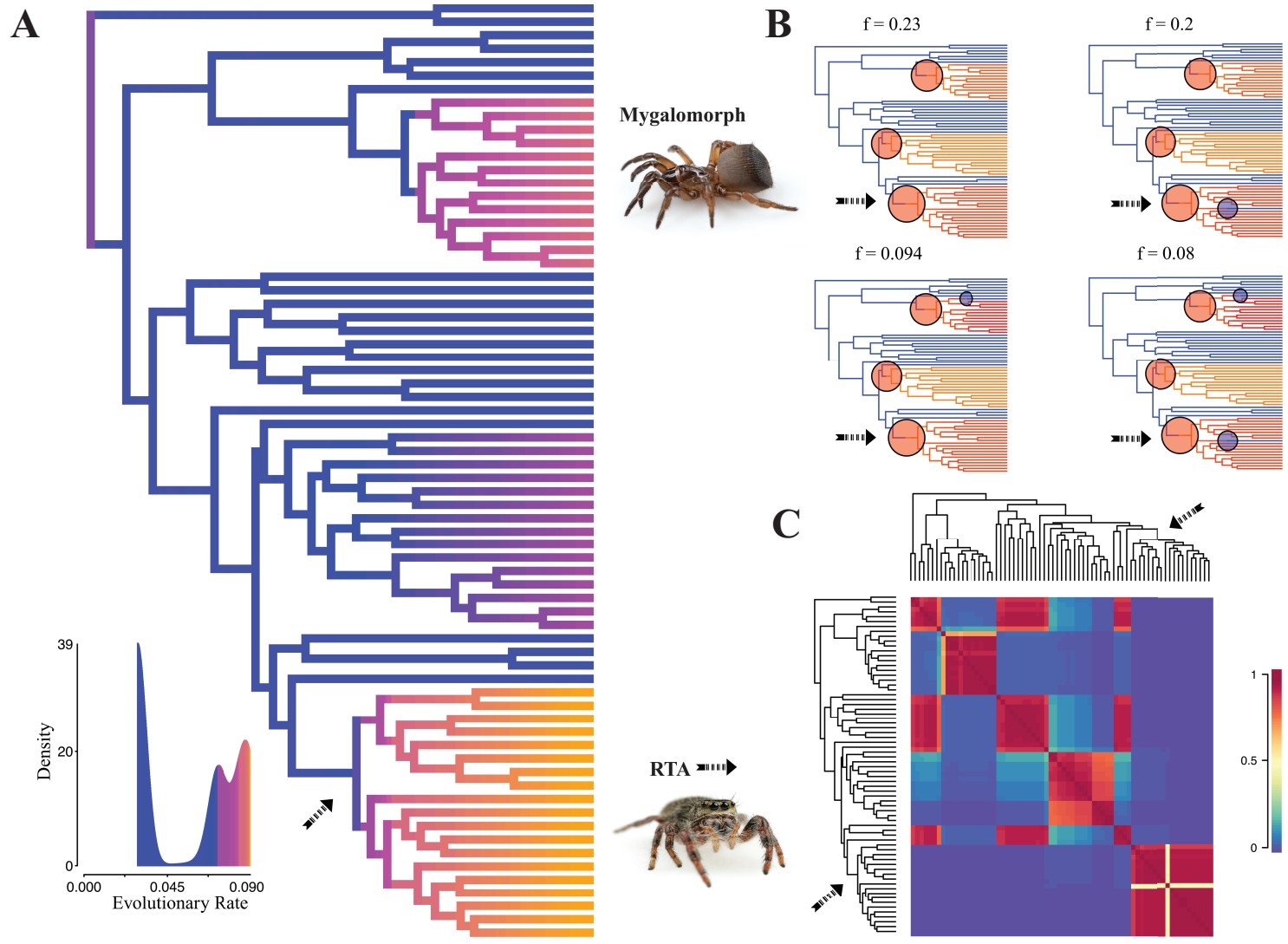

**Figure 5** Time-calibrated phylogeny of spiders with branches colored by reconstructed net diversification rates (A). Rates on branches are means of the marginal densities of branch-specific rates. Inset histogram shows posterior density of speciation rates. Smaller phylogenies (B) show the four distinct shift configurations with the highest posterior probability. For each distinct shift configuration, the locations of rate shifts are shown as red (rate increases) and blue (rate decreases) circles, with circle size proportional to the marginal probability of the shift. The macroevolutionary cohort analysis (C) displays the pairwise probability that any two species share a common macroevolutionary rate dynamic. Dashed arrow indicates position of RTA clade on each tree.

the orb web. It shows that the Haplogynae are polyphyletic with Filistatidae as sister to Hypochilidae and Leptonetidae as sister to Entelegynae. It also suggests a position for two enigmatic families—Hahniidae and Homalonychidae—and provides an alternate view of RTA relationships and the contents of Dionycha clade.

## Data characteristics and development of spider core orthologs

Transcriptome analyses are unquestionably data rich. Thousands of assembled sequences emerge from even modest RNA-seq experiments, providing, among other things, a basis for identifying phylogenetically informative orthologs. This bounty comes with a few

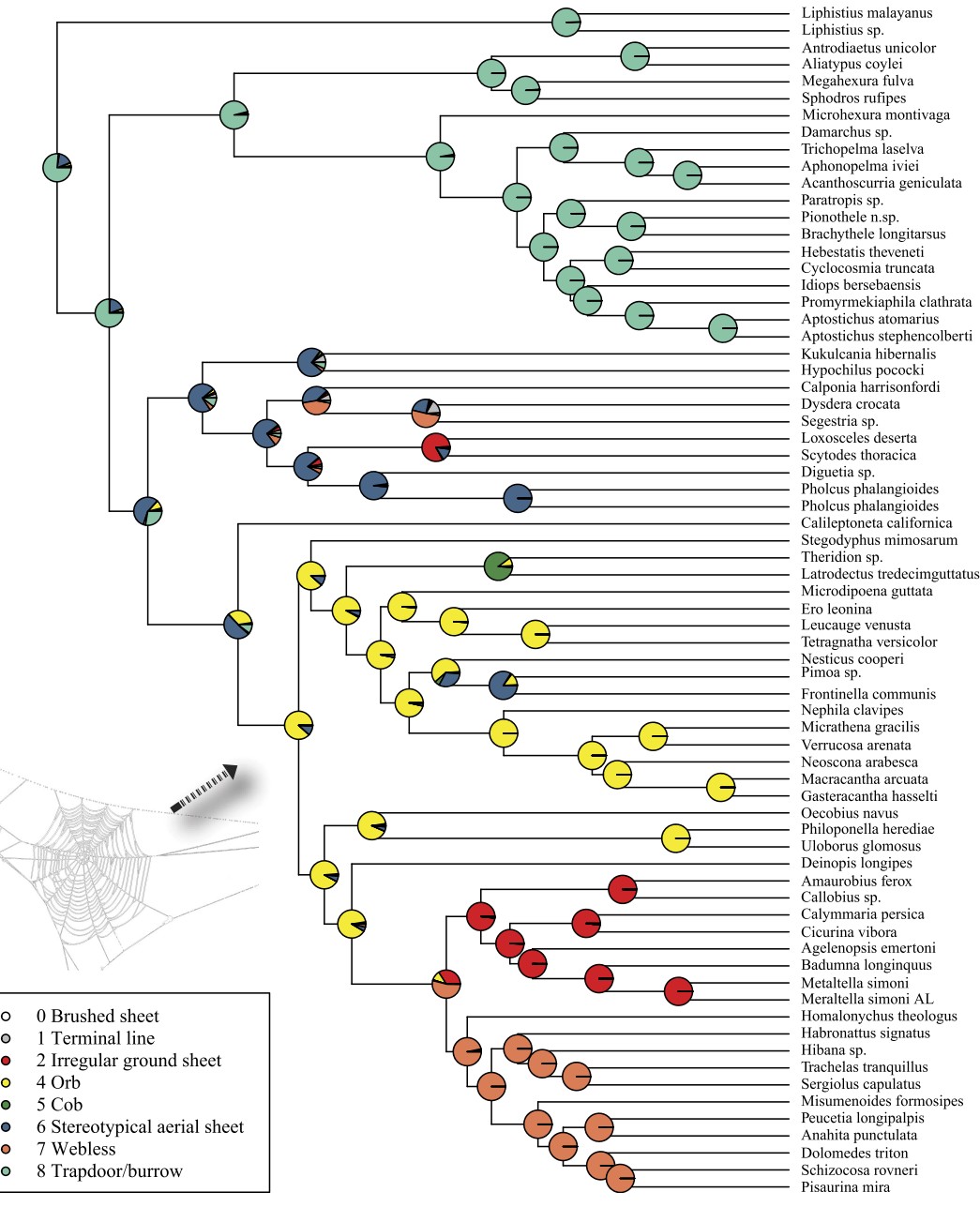

**Figure 6  ML ancestral state reconstructions of web type on the time-calibrated phylogeny of spiders.** Circle areas correspond to probability of ancestral states. The arrow points to the origin of the orb web at the MRCA of Entelegynae excluding Leptonetidae.

caveats. Isoforms, paralogous sequences, and assembly artifacts (chimeric contigs) can mislead inference of single-copy orthologous genes. The data represent one snapshot—a specific organism, point in time, and combination of tissues—that can lead to gaps in downstream supermatrices due to stochastic sampling issues. Large amounts of missing data, due to missing loci and indels introduced during alignment, can arise post-assembly

in the ortholog detection and filtering stages of phylogenomic analyses (*Bond et al., 2014*; *Fernández, Hormiga & Giribet, 2014*). *Lemmon et al. (2009)* and a number of other authors (*Roure, Baurain & Philippe, 2013*; *Dell'Ampio et al., 2014*; *Xia, 2014*) have discussed the potential negative effects of such missing data in large phylogenomic (transcriptome-based) datasets. Recent studies argue that the phylogenetic signal from transcriptomes can conflict with alternative reduced representation approaches like targeted sequence capture (*Jarvis et al., 2014*; *Brandley et al., 2015*; *Prum et al., 2015*). From vast amounts of bird genome protein-coding data, *Jarvis et al. (2014)* concluded that these loci were not only insufficient (low support values), but also misleading due to convergence and high levels of incomplete lineage sorting during rapid radiations.

Simulation studies now predict that 10s–100s of loci will resolve most phylogenies, albeit this is sensitive to factors such as population size or speciation tempos (*Knowles & Kubatko, 2011*; *Leache & Rannala, 2011*; *Liu & Yu, 2011*). To mitigate the impacts of paralogy, incomplete lineage sorting, and missing data, we developed *a priori* a set of spider core orthologs that comprise a database consisting of over 4,500 genes that are expected to be recovered from most whole spider RNA extractions and are likely orthologous. We summarize the annotations for each of the genes in the HaMStR pHMM file in Table S3.

Our approach enhances repeatability, downstream assessment, scalability (taxon addition), and data quality. Studies that employ pure clustering approaches like OMA stand-alone (*Altenhoff et al., 2013*) may produce more data (i.e., more "genes") on the front end; however, they present some problems in terms of ease of scalability. Although adding more genes is one strategy, it is increasingly clear that taxon sampling and data quality are very important (*Lemmon & Lemmon, 2013*; *Bond et al., 2014*).

## A modified view of spider evolution and key innovations

Once considered the "crowning achievement of aerial spiders" (*Gertsch, 1979*), the orb web and consequent adaptive radiation of araneoid spiders (ecribellate orb weavers and their relatives) has captured the imagination of spider researchers for over a century. The evolution of adhesive threads and the vertical orientation of the orb web, positioned to intercept and retain flying insects, has been long considered a "key innovation" that allowed spiders to inhabit a new adaptive zone (*Bond & Opell, 1998*). It is important to note that several prior authors speculated about orb web adaptive value, such as *Levi (1980)*, *Opell (1979)*, *Opell (1982)* and *Coddington (1986)* although *Bond & Opell (1998)* quantified the pattern in a formal phylogenetic framework. Over 25% of all described spider species are araneoids. Given orb weaver monophyly on quantitative phylogenies (*Griswold et al., 1998*; *Blackledge et al., 2009*), rigorous empirical studies tended to confirm the orb as a prime cause of spider diversification (*Bond & Opell, 1998*). Nevertheless, a lack of correlation of the orb web and species richness has been apparent for some time. *Griswold et al. (1998)* noted that over 50% of Araneoidea no longer build recognizable orb webs and suggested that "the orb web has been an evolutionary base camp rather than a summit."

*Bond et al. (2014)* tested two alternative evolutionary scenarios for orb web evolution, reflecting different analytical results; parsimony implied multiple independent origins, and maximum likelihood implied one origin and subsequent multiple losses. The current

study (Fig. 6) favors the latter: the orb evolves at the base of the araneoid + deinopoid + RTA clade, but is lost at least three times independently. Large amounts of morphological and behavioral data (albeit often correlated with features essential to the orb) still support the single origin hypothesis (*Coddington, 1986*; *Coddington, 1991*; *Scharff & Coddington, 1997*; *Griswold et al., 1998*; *Agnarsson, Coddington & Kuntner, 2013*). Our results suggest both that the orb web originated earlier than previously supposed, and that heretofore-unsuspected clades of spiders descend from orb weavers. In a sense, this ancient origin hypothesis reconciles the implications of genomic data with the classical evidence for multiple, homologous, complex, co-adapted character systems.

Recent discoveries of large, cribellate orb web-weaving taxa from the late Triassic agree with our molecular dates. Diverse Mesozoic deinopoids (*Selden, Ren & Shih, 2015*) are consistent with the "orb web node" at 213 Ma (Fig. 4 and Table 3). Under this view, modern uloborids and deinopids are distinct remnants of this diverse group. *Selden, Ren & Shih (2015)* previously noted that if other extant taxa "emerged from the deinopoid stem or crown group it would render the whole-group Deinopoidea paraphyletic"; we discuss this scenario in detail below.

Contrary to the contemporary paradigm that the evolution of the orb web and adhesive sticky threads elevated rates of diversification among the araneoid spiders, our BAMM analysis (Fig. 5) indicates that the highest rates of diversification likely occurred among the RTA spiders followed by mygalomorphs and then araneoids as a distant third, the latter driven–in part–by the secondarily non-orb weaving theridiids and linyphiids. These results imply that other foraging strategies (e.g., cursorial hunting and irregular sheets) were a more "successful" strategy than the orb. Indeed, the point estimate for the RTA node during the early Cretaceous (138.8 Ma; Fig. 4 and Table 3) precedes the subsequent diversification of the RTA clade at 125–100 Ma.

This date coincides with the Cretaceous Terrestrial Revolution (KTR). Angiosperms radiated extensively at 125–90 Ma (*Crane & Friis, 1987*; *Wang, Zhang & Jarzembowski, 2013*), as did various plant-dependent insect lineages, including beetles (*McKenna et al., 2009*; *Mckenna et al., 2015*), lepidopterans (*Wahlberg, Wheat & Peña, 2013*), ants (*Moreau et al., 2006*) and holometabolous insects in general (*Misof et al., 2014*), although some insect lineages do not show a pulse (e.g., darkling beetles; *Kergoat et al., 2014*). Spiders, as important insect predators, may also have diversified rapidly along with their prey (e.g., *Penney, Wheater & Selden, 2003*; *Peñalver, 2006*; *Selden & Penney, 2010*). The fossil and phylogenomic data presented here show that most spider lineages predate the KTR (*Selden & Penney, 2010*; *Bond et al., 2014*). Among these, the RTA clade especially, but also mygalomorphs and araneoids, diversified in response to the KTR insect pulse. That aerial web spinners specialized on rapidly radiating clades of flying insects is hardly surprising. Similarly, if forest litter habitats became more complex and spurred insect diversification (*Moreau et al., 2006*) ground-dwelling spiders may also have diversified at unusual rates. Perhaps the most dramatic change in insect abundances occurred with the origin and early diversification of social insects (*Hölldobler & Wilson, 1990*) and beetles (*Mckenna et al., 2015*). Both groups

date back to 150–125 Ma and diversified during the KTR (*LaPolla, Dlussky & Perrichot, 2013*; *Ward, 2014*; *Legendre et al., 2015*). A major increase in these insect groups may have favoured spiders that feed on cursorial prey and thus could help explain the concurrent increase in diversification in the RTA clade, mygalomorphs, and non-orb weaving araneoids such as cobweb weavers (*Dziki et al., 2015*).

Taken together, this new evidence on character evolution, divergence estimates, and rates of diversification indicates that previous conclusions regarding the timing and rate of spider evolution were imprecise. Our data support an ancient orb web hypothesis that is further bolstered by a wealth of fossil data showing that a cribellate deinopoid stem group likely diversified during the early Mesozoic. Molecular divergence clock estimates are consistent with the placement of the orb web further down the tree as well as suggesting that some of the greatest rates of species diversification coincided with the KTR. The latter suggests that spiders took advantage of increased abundance of cursorial prey.

These findings likely diminish the hypothesis proposed by *Bond & Opell (1998)* that the vertically oriented orb web represented a key innovation, particularly in light of the fact that over half of araneoid species do not build an orb web (e.g. Theridiidae and Linyphiidae; noted by *Griswold et al., 1998*; *Fernández, Hormiga & Giribet, 2014*). We already knew that major orb web-weaving groups are very successful in spite of abandoning the orb (*Blackledge et al., 2009*).

## Spider systematics

Although our results show that many classical ideas in spider systematics require revision (e.g., mygalomorph families, Haplogynae, paleocribellates, higher araneoids, and RTA + dionychan lineages), they also robustly support many classical taxonomic concepts.

### *Mygalomorphae relationships*

Since *Raven (1985)*, Mygalomorphae (Table 1, Node 4) has continuously represented a challenge to spider systematics. As discussed by *Hedin & Bond (2006)* and *Bond et al. (2012)*, nearly half the families are probably non-monophyletic. While our sampling here and previously (*Bond et al., 2014*) is far greater than any other published phylogenomic study (e.g., *Fernández, Hormiga & Giribet (2014)* included just one theraphosid), taxon sampling remains insufficient to address major issues aside from deeper level phylogenetic problems. However, the data (Fig. 2) support Euctenizidae as a monophyletic family, but not Nemesiidae. As indicated in *Bond et al. (2014)*, the once controversial Atypoidina (Node 5) consistently has strong statistical support in all analyses. Alternatively, the placement of paratropidids, ctenizids, and idiopids remains questionable and warrants further sampling.

### *Haplogynae relationships*

The traditional view of spider classification (*Coddington, 2005*) places Paleocribellatae and Austrochiloidea (Table 1) as sister groups to all the remaining Araneomorphae taxa—Haplogynae and Entelegynae; the latter terms are used primarily herein as clade names rather than specific reference to genitalic condition. Our current tree (Fig. 2) is congruent with *Bond et al. (2014)* in placing Paleocribellatae (Table 1, *Hypochilus*; Fig. 1,

Node 9) as sister to Haplogynae. Filistatidae (*Kukulcania*), which is placed as sister to the ecribellate haplogynes (Synspermiata lineage as proposed in *Michalik & Ramírez, 2014*), pairs with *Hypochilus* as in *Bond et al. (2014)*. This arrangement suggests that characters formerly considered "primitive" to araneomorphs, for example, mobile leg three cribellate silk carding, might instead be a synapomorphy for the new hypochilid-filistatid clade. Remaining haplogyne relationships are somewhat congruent with previously published analyses (*Ramírez, 2000*; *Michalik & Ramírez, 2014*). However, one of the more intriguing results is the placement of the morphologically intermediate "haplogyne" (Table 1) *Calileptoneta* (Leptonetidae) as sister to Entelegynae, suggesting that leptonetids may represent intermediate genitalic forms between haplogyne and the relatively more complex entelegyne condition (*Ledford & Griswold, 2010*). As outlined by *Ledford & Griswold (2010)*, a number of previous analyses (*Platnick et al., 1991*; *Ramírez, 2000*; *Griswold et al., 2005*) discussed the "rampant" homoplasy required to place leptonetids (sister to Telemidae) among haplogynes and suggest two possible scenarios—leptonetids are proto-entelegynes, or they are the sister group to the remaining Haplogynae. Our phylogenomic analyses support the former hypothesis favored by *Ledford & Griswold (2010)*, and puts the discovery of the cribellate *Archoleptoneta* into better phylogenetic context. Additionally, these results provide further support for the concept of Synspermiata as proposed by *Michalik & Ramírez (2014)* and represent a robust phylogenetic framework for understanding the evolution of entelegyne genitalia.

### Araneoidea relationships

Our reconstruction of araneoid relationships departs dramatically from the traditional classification scheme and a number of recently published molecular systematic studies (e.g., *Blackledge et al., 2009*; *Dimitrov et al., 2012*). Theridiidae (cobweb spiders) is sister to the remaining araneoids as opposed to occupying a more derived position within that clade. Comparisons to *Dimitrov et al. (2012)* should be viewed with caution: that analysis contained a large suite of taxa not included here, and many results of that analysis had only weak support. Nevertheless, our phylogenomic data agree in supporting the close relationship between Mysmenidae, Mimetidae, and Tetragnathidae. We also retain the more inclusive linyphioids as close relatives of Araneidae + Nephilidae as in *Dimitrov et al. (2012)*. Unlike that study, we recover nesticids sister to linyphioids (Pimoidae plus Linyphiidae) rather than theridiids: Theridioid (Theridiidae and Nesticidae) diphyly is a surprising result, which has already been shown with standard markers by *Agnarsson, Coddington & Kuntner (2013)*. Theridioids have strikingly similar spinning organs and tarsus IV comb for throwing silk, but are otherwise genitalically distinct. Clearly relationships among the derived araneoids require more intensive sampling, especially of missing families (Theridiosomatidae, Malkaridae, Anapidae, etc.) to adequately resolve their phylogeny.

### Deinopoidea relationships

The addition of nearly 30 terminals to the *Bond et al. (2014)* dataset corroborates the non-monophyly of the classically defined Orbiculariae, although the orb and its behavioral, morphological, and structural constituents may be homologous. Deinopoidea, with these

data, is polyphyletic (see also *Dimitrov et al., 2012*). Instead, a new clade, Uloboridae + Oecobiidae, is sister to Deinopidae + the RTA clade. Bootstrap support was consistently low for the node dividing these two groupings in all analyses except matrix 6 (Fig. 2), which omits the eresid exemplar *Stegodyphus* and matrix 8, the ASTRAL analysis. The placement of the two eresoid taxa (Table 1), *Stegodyphus* and *Oecobius* continues to present difficulties here as in previous published phylogenomic studies (*Miller et al., 2010*). *Fernández, Hormiga & Giribet (2014)* found alternative placements for *Oecobius* (their only eresoid) whereas *Bond et al. (2014)* typically recovered *Stegodyphus* as the sister group to all entelegynes (recovered here as the sister group to araneoids) and *Oecobius* as a member of a clade comprising uloborid and deinopid exemplars, but with notably lower support. Disparities between the two analyses may be attributed to differences in taxon sampling, which, as noted above, was far greater in *Bond et al. (2014)*. On the other hand, increased taxon sampling across the tree diminished node support in some places. However, it is worth noting that support was very strong in the ASTRAL species tree analysis, suggesting that while there may be some conflict among individual data partitions there is an overwhelming amount of signal in the data for a Deinopoidea + RTA relationship. This trend was noted by *Bond et al. (2014)* who found that only 2.4% of all bootstrap replicates recovered a monophyletic Orbiculariae. Based on these data and the putative rapid diversification that occurred once the orb web was abandoned, it is clear that resolving relationships at this point in spider evolutionary history remains a challenge. Finally, *Bond et al. (2014)* and *Agnarsson, Coddington & Kuntner (2013)* recovered an unexpected relationship between eresoid taxa and deinopoids that consistently rendered the Deinopoidea paraphyletic or polyphyletic if *Oecobius* was included in the analysis. Our results, here including an additional uloborid exemplar, still confirm Deinopoidea polyphyly. Perhaps careful examination of *Oecobius* web morphology and spinning behavior will provide independent corroboration of this molecular signal.

### RTA and Dionycha relationships

Although all of our analyses recover a monophyletic RTA clade, relationships among its members reflect some departure from the traditional view of RTA phylogeny but are largely consistent with a more recent morphology-based study. We recover a clade that comprises a mix of agelenoids (Agelenidae, Desidae, and Amphinectidae) as a sister group to Dictynidae + Hahniidae and Amaurobiidae. The taxonomic composition of Dictynidae, Hahniidae and Amaurobiidae, as well as their phylogenetic placement, remains problematic and in a state of flux (*Coddington, 2005*; *Spagna, Crews & Gillespie, 2010*; *Miller et al., 2010*). The typical hahniines have been difficult to place due to their long branches (*Spagna & Gillespie, 2008*; *Miller et al., 2010*). *Calymmaria*, has been moved into "Cybaeidae s.l." by *Spagna, Crews & Gillespie (2010)*, suggesting that the relationships among hahniids, cybaeids, and dictynids need further scrutiny.

Amaurobiids have also been hard to place, though this is in part because Amaurobiidae are a moving target. The term Amaurobiids needs to be clarified, as most of nine subfamilies discussed in *Lehtinen (1967)* are now placed elsewhere. We use *Callobius*, from the type subfamily of the family. Our amaurobiid placement, basal to an agelenoid and dictynoid

grouping corroborates previous findings (*Miller et al., 2010*; *Spagna, Crews & Gillespie, 2010*). Dictynids on the other hand were considered one of the unresolved sister groups to amaurobioids, zodarioids, and dionychans (*Spagna, Crews & Gillespie, 2010*). Here the placement of our dictynid exemplar *Cicurina* is more precise: sister group to the hahniid *Calymmaria* (as in *Miller et al., 2010*).

We also recover Homalonychidae (representing Zodarioidea) as the sister group to dionychans and lycosoids, once again, mirroring the results of *Agnarsson, Coddington & Kuntner (2013)*. Previously Zodarioidea was placed closer to the base of the RTA clade (*Miller et al., 2010*). Dionychans here include salticids, anyphaenids, corinnids, and gnaphosids whereas crab spiders (Thomisidae) nest with the lycosoids containing a paraphyletic Pisauridae. Placement of Thomisidae within Lycosoidea goes back at least to *Homann (1971)* and was formally established by *Bayer & Schönhofer (2013)* and the total evidence analysis of *Polotow, Carmichael & Griswold (2015)*. Although *Ramírez (2014)* placed Thomisidae outside of Lycosoidea, in one of his slightly suboptimal results thomisids were included in Lycosoidea. The relationships we recover among dionychan and lycosoid taxa are largely congruent with those inferred by *Ramírez (2014)* in a massive morphological study of Dionycha and RTA exemplars. Given the general incongruence among previous morphological and molecular spider systematic studies, it will be interesting to see how *Ramírez (2014)* phylogeny and familial-level reevaluations compare as phylogenomic studies expand. *Raven (1985)* was a landmark study for mygalomorphs; perhaps *Ramírez (2014)* may serve in the same capacity for one of the most diverse branches on the spider tree of life.

## CONCLUSIONS

Following *Coddington & Levi (1991)*, higher-level spider classification underwent a series of challenges from quantitative studies of morphology, producing provocative but weakly-supported hypotheses (*Griswold et al., 1998*; *Griswold et al., 2005*). Total evidence studies, for example, *Wood, Griswold & Gillespie (2012a)*; *Wood et al. (2012b)* for Palpimanoidea, *Polotow, Carmichael & Griswold (2015)* for Lycosoidea, and *Bond et al. (2012)* for Mygalomorphae appear to have settled some local arrangements, but much of the backbone of the spider tree of life remains an open question only to be solved through increased taxon sampling. Phylogenomics has already brought data-rich, convincing solutions to long standing controversies, for example, phylogeny of the orb web (*Bond et al., 2014*; *Fernández, Hormiga & Giribet, 2014*). Phylogenomics portends a new and exciting period for spider evolutionary biology. Recent advances in digital imaging, proteomics, silk biology and major fossil discoveries mean that our understanding of spider evolution will likely accelerate by leaps and bounds in the coming years. The tempo and mode of spider evolution is likely different than previously thought. At this point it seems reasonably clear that the orb web evolved earlier phylogenetically than previously thought, only to be subsequently lost at least three times independently during the Cretaceous. While the orb web has certainly been successful, a likely dramatic increase in the abundances of cursorial insects during the KTR also impacted the success of other foraging strategies, including

webless hunting. Our results and those of others like *Ramírez (2014)* show that spider systematics still remains a work in progress with many questions yet unanswered.

## ACKNOWLEDGEMENTS

This is contribution 730 of the Auburn University Museum of Natural History. The authors would like to thank an anonymous reviewer, S Edwards, F Labarque, P Michalik, J Miller, MJ Ramirez, and R Raven for insightful comments on earlier drafts of this manuscript.

### Funding

This work was partially funded by Auburn University and National Science Foundation grant DEB 1256139. The funders had no role in study design, data collection and analysis, decision to publish, or preparation of the manuscript.

### Grant Disclosures

The following grant information was disclosed by the authors:
Auburn University and National Science Foundation grant: DEB 1256139.

### Competing Interests

The authors declare there are no competing interests.

### Author Contributions

- Nicole L. Garrison conceived and designed the experiments, performed the experiments, analyzed the data, wrote the paper, prepared figures and/or tables, reviewed drafts of the paper.
- Juanita Rodriguez and Kevin M. Kocot conceived and designed the experiments, performed the experiments, analyzed the data, wrote the paper, reviewed drafts of the paper.
- Ingi Agnarsson reviewed drafts of the paper.
- Jonathan A. Coddington wrote the paper, reviewed drafts of the paper.
- Charles E. Griswold and Marshal Hedin contributed reagents/materials/analysis tools, wrote the paper, reviewed drafts of the paper.
- Christopher A. Hamilton analyzed the data, reviewed drafts of the paper.
- Joel M. Ledford contributed reagents/materials/analysis tools, reviewed drafts of the paper.
- Jason E. Bond conceived and designed the experiments, performed the experiments, analyzed the data, contributed reagents/materials/analysis tools, wrote the paper, prepared figures and/or tables, reviewed drafts of the paper.

## DNA Deposition

The following information was supplied regarding the deposition of DNA sequences:

Illumina transcriptome sequence data are available from NCBI database archive under accession numbers SAMN04453329–SAMN04453350. Phylogenomics data matrices were deposited on 5 February 2016 in the Dryad Digital Repository at doi:10.5061/dryad.6p072

## Data Availability

Dryad data doi:10.5061/dryad.6p072.

## Supplemental Information

Supplemental information for this article can be found online at http://dx.doi.org/10.7717/peerj.1719#supplemental-information.

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
