# Peer review of "Spider phylogenomics: untangling the Spider Tree of Life"

_PeerJ, doi:10.7717/peerj.1719_

## Round 0.1 · original submission · Major Revisions

· Academic Editor

Major Revisions

I have received two very thorough reviews. Both reviewers feel the paper adds important new data to an important problem. But both also feel that the paper could do a better job of placing itself in the context of recent and ongoing work on spider phylogenomics. Reviewer 1 clearly has many comments devoted to particular citations and how your results are cast vis a vis previous work. I encourage you to look carefully at each of these suggestions.

Reviewer 2 felt the paper was very strong but again mentioned a clear need to present the work in a more balanced way, with less emphasis on perceived superiority of the present work compared to other recent studies. In general I think it is best to consider this work as a work in progress for spider phylogenomics that may suffer from some shortcomings in ways similar to previous studies. Undoubtedly it is an advance for the field but please try to present a more balanced view of the work as compared to other recent studies.

I note that you use the ASTRAL approach. This is great, although I would refrain from calling it a "coalescent" approach. The clustering algorithm in ASTRAL is not based on coalescent theory, although it takes its justification from a singularity about quartets that is based in coalescent theory. I would be happy to talk to you more about this if you want, but otherwise please see a recent review in MPE (Edwards et al. 2015: http://www.sciencedirect.com/science/article/pii/S1055790315003309) I don't mean to promote my own work here but another statement that you make on page 5, that "The ASTRAL approach is thought to be more robust to incomplete lineage sorting, or deep coalescence, than concatenation methods or other shortcut coalescent-based approaches" is also not true, or at the very least is an exaggeration of its performance compared to other true coalescent methods. Again, I don't mean to promote my own biases, but I believe it's best thought of as a fast alternative, rather than a superior approach compared to other "shortcut" methods. I also discourage you (but of course don't require you) from using the term "shortcut" coalescent method. I would prefer you use the term "two-step" coalescent method (again, see above review and references therein). Finally, the statement that "Phylogeny inference may also be misled by recombination (Springer and Gatesy, 2016)" is also not true, and certainly has not been demonstrated. Our MPE review discusses all of these issues, and your paper is an important contribution to the ongoing debates.

Overall your paper has many merits although it may require additional review before a final decision can be made. Thank you for submitting to PeerJ.

·

Basic reporting

1) The article should include sufficient introduction and background to demonstrate how the work fits into the broader field of knowledge. Relevant prior literature should be appropriately referenced.
In the Introduction, line 31, the authors present several reviews references, e.g. Agnarsson et al. 2013 and Bond and Opell, 1998. I think is relevant to provide the most important references in those specific areas, and then provide the reviews/summaries of them.
Also, line 81, the authors write “Notably, Bond et al. (2014) rejected Orbiculariae, […]” This is also recalled in Fernández et al. (2014: fig. 2 A, B).

2) Figures should be relevant to the content of the article, of sufficient resolution, and appropriately described and labeled.
In figure 6, page 28, the circles are too small with stroke dark lines (see option in Photoshop). This may confuse the readers as the authors used also black color for their legend "1 Terminal Line". Also, the arrow in figure 6 is pointing to the Entelegynae clade (excluding Leptonetidae); instead to the RTA clade (see figure legend).

Experimental design

No comments.

Validity of the findings

1) The conclusions should be appropriately stated, should be connected to the original question investigated, and should be limited to those supported by the results.
In this manuscript, thus, the conclusions should be frame particularly in the Spider Tree of Life and generally in Phylogenomics. Regarding the first, the manuscript presents several inconsistencies in the cited references (e.g. not citing relevant literature, citing quotes that actually are not in the cited papers), on the systematic treatment of novel phylogenetic hypotheses (e.g. overlooking valid monophyletic clades as Synspermiata, not discussing potential/plausible alternative phylogenetic hypotheses as the putative placement of Austrochiloidea). Here I summarize the main issues regarding those topics:
Synspermiata (line 304, also see line 458)
The authors results (Figure 2) show that the clades of Filistatidae plus Hypochiliidae, Synspermiata and ((Filistatidae, Hypochiliidae) Synspermiata)) are well supported.
The term Haplogynae must be avoided for naming a monophyletic clade, as there are several examples of species with a female haplogynae condition that emerge independently in different parts of the spider tree of life, e.g., some genera of Tetragnathidae, Uloboridae, Anapidae and Palpimanoidea, and in this study Leptonetidae.
This idea is also strengthened by the recently erected Synspermiata clade. Sysnpermiata clustered all the classical haplogynae families except Filistatidae (Michalick and Ramírez 2014: fig. 9). It is also exclude Hypochiliidae, which is in concordance with the authors results (Fig. 2). Even more, the splitting of Leptonetidae to become the sister group of the rest of Entelegynae spiders (Fig. 2) do not invalidate Synspermiata, as Protoleptoneta italica (the only leptonetid tested in their study) present cleistospermia (common for most Araneomorphae). Thus, Synspermiata is still a valid monophyletic clade and must be used.
However, the authors are welcome to coin a new or raise an old name for the clade ((Filistatidae, Hypochiliidae) Synspermiata)), which is well supported in all the analyses.

Filistatidae + Hipochillidae (line 306; also see line 342)
The authors suggest that “Filistatidae (Kukulcania) is removed from other haplogynes and emerges as the sister group to Hypochilus.” I will suggest rather the opposite: that Hypochilus (which was classically considered the sister group of the rest of Araneomorphae: Paleocribellatae) emerges as the sister group of Filistatidae (which classically was the sister group of the classical haplogynae). The authors results (Fig. 2) show that Hypochilus plus Filistatidae are the sister group of Synspermiata (the old classical haplogynae excluding Filistatidae and, now, also Leptonetidae).

Eresidae (line 308)
Eresidae is low supported in the authors results (Fig. 2), boostrap proportion (BS) less than 0.7, except, in scheme 8 (BS 0.9). As they are showing well-supported clades in this section, I will suggest adding these results in another paragraph or just named it in the Discussion.
Also, Eresidae was already suggested as sister group of Araneoidea by Miller et al. 2010 (fig. 3). Similar to the authors’ results, this clade was recovered with low support by Miller et al. 2010. Thus, it is not necessary to emphasized that the position of Eresidae is controversial. However, the authors should mention the previous hypothesis of Miller et al. (2010).

Deinopoidea + Oecobiidae (line 308)
The relations between Oecobiidae, Uloboridae and Deinopidae (the last two classically forming Deinopoidea) are inconclusive in the authors results (Fig. 2), as reflected by the low boostrap proportions (less than 0.7). However, in Figure 3, those clades are well supported. In the other hand, their results show high support in the node clustering those groups with the RTA clade, using the schemes 5 to 8. As before, I will recommend showing these results in another paragraph as the authors are showing here a summary of all the well-supported clades in all analyses.
Also, Deinopoidea had emerge polyphyletic in previous analyses (e.g. Dimitrov et al. 2012), also with low support as in the authors results.

Chronogram: Figure 4 (line 322)
I wander if the authors explore the diversification rate shifts using a matrix that includes Calileptoneta. It would be really interesting, even desired, to estimate the age of the node that cluster Calileptoneta with Entelegynae, as this is one of the authors’ novel and shocking results.

Data Characteristics and Development of Spider Core Orthologs (line 374)
Here I'm also wandering why the authors add Lopardo and Hormiga (2015)’ reference in this section.
In this section, the authors are highlighting the importance of enhance both collection and analyses of genomic data. All references in the paragraph apply to this idea except by the work of Lopardo and Hormiga (2015), which deals essentially with morphological characters (e.g. description of characters and states, phylogeny analyses using those characters) and mention briefly previous total evidence phylogenetic analyses (morphological plus molecular data in Lopardo et al. 2011). Maybe, the authors actually want to refer to Lopardo et al. (2011). But note that those authors use only 6 genes in their analyses. What is the meaning of the following statement: "adding more genes is one strategy (e.g. Lopardo and Hormiga, 2015)"? Clearly, the approaches, focuses and methodologies of these mentioned papers were totally different from doing phylogenomics.
Furthermore, please see the complete sentence: "Although adding more genes is one strategy (e.g., (Lopardo and Hormiga, 2015), it is increasingly clear that taxon sampling and data quality are more important than quantity (Lemmon and Lemmon, 2013; Bond et al., 2014)." In my opinion, this sentence seems biased. A naive reader (e.g. not familiar with the arachnological literature and their authors) may think that taxon sampling and data quality in Lopardo and Hormiga (2015) were poor, when is not the case. Their matrix consists of 357 morphological characters and 70 terminal taxa (e.g. 47 ingroups plus 23 outgroups). Thus, I suggest deleting this reference from the entire paragraph, or rewrite it in another way.

A Modified View of Spider Evolution and Key Innovations: Figure 6 (line 392)
I wander if the authors explore both ancestral state reconstructions analyses (e.g. parsimony vs. maximum likelihood –ML–) for their data. Figure 6 only shows the ML result. It would be interesting to show both approaches, especially if their have differences, as in Bond et al. (2014).

Diverse Mesozoic deinopoids (line 401, also see line 435)
I wonder why the authors mention "Diverse Mesozoic deinopoids (Selden et al., 2015)", when in fact that paper do not deal with the description of Mesozoic deinopoids but with the transfer of a presumably araneoids fossils to Opiliones. From Selden et al. (2015): "Here, we focus on a case study involving some putative spiders (Arachnida: Araneae) described by Pocock (1911) and Petrunkevitch (1949) from the Coal Measures of Coseley in the English West Midlands (Figs 1-5). The work of these authors implied a Palaeozoic record of spiders belonging to the derived infraorder Araneomorphae: the fossils were likened to modern orb-weaving spider genera such as Nephila, Tetragnatha or Deinopis. With the help of computed microtomography (mCT), we demonstrate that at least one of these fossils is an unusual, long-bodied harvestman (Arachnida: Opiliones). Thus, the species of Pocock and Petrunkevitch can be formally excluded from the spider fossil record [emphazis added]."
Maybe, the authors want to refer to Selden (1990): "Spiders are rare in rocks of Mesozoic age. [...] The four specimens described here are sufficiently well preserved for their taxonomic affinities to be determined with some precision, and thus they represent only the third find of Mesozoic spiders to be described and named. The fossil spiders described here are placed in extant superfamilies or families." The authors may see that there is a unique fossil described for Uloboridae: Palaeouloborus lacasae. So, I wander why they also write "Diverse Mesozoic deinopoids". The same information can be found in Selden and Penney (2003): "Selden (1990) suggested that, of the two deinopoid families Palaeouloborus was closer to Uloboridae than Deinopidae." From Selden et al. (1999): "Few Mesozoic spiders have been described. [...] orbicularian araneomorphs (Selden 1990; Mesquita 1996). [...] No other fossil araneomorph spiders are known from the Paleozoic era or the Triassic period of the Mesozoic era, thus the specimens described herein are the oldest known fossil spiders which can be referred to Araneomorphae [emphazis added] with some degree of confidence." "The discovery of araneomorph spiders in the Triassic period is not unexpected, since the existence of their sister group, Mygalomorphae, in strata of similar age (Selden & Gall 1992) predicts this." Here Selden et al. described Triassaraneus andersonorum presumably an Araneidae. However, this interpretation changes in 2009.
Or, maybe, the authors want to cite the figure 1 in Selden et al. (2009) and compare their results with it. Triassaraneus andersonorum, in that paper, is suggested as the sister group of Araneomorphae. The fossil records of the classic Deinopoidea (Uloboridae plus Deinopidae) date from 145 mya. This actually agrees with the authors suggestion of the emerging “orb web node” at 213 mya. From Selden et al. (2009): "Species from the following spider families are currently known as sub-fossils in Madagascan copal: Araneidae, Archaeidae, Clubionidae, Corinnidae, Deinopidae [emphazis added], Dictynidae, Hahniidae, Hersiliidae, Linyphiidae, ?Migidae, ?Miturgidae, Mysmenidae, Oonopidae, Philodromidae, Pholcidae, Salticidae, Scytodidae, Segestriidae, Selenopidae, Tetragnathidae, Theridiidae, Thomisidae and Uloboridae (Wunderlich 2004, 2008)."

Diverse Mesozoic deinopoids (line 403)
The authors should check Selden et al. 2015, there is not such a line like "emerged from the deinopoid stem or crown group it would render the whole-group Deinopoidea paraphyletic". Here is the reference to that paper:
Selden, P. A., Dunlop, J. A., and Garwood, R. J. (2015). Carboniferous araneomorph spiders reinterpreted as long-bodied harvestmen. Journal of Systematic Palaeontology, (ahead-of-print):1–11.

Paleocribellatae + Austrochiloidea (line 459 and 462)
Following Coddington (2005: fig. 2.2), Paleocribellatae is the sister group of Neocribellatae that includes Austrochiloidea as sister group of Araneoclada (Haplogynae plus Entelegynae). In other words, the relations should be written like this: (Hypochillidae (Austrochiloidea (Haplogynae, Entelegynae))). The authors wrote that “[…] Paleocribellatae and Austrochiloidea (Table 1) as sister groups to all the remaining Araneomorphae […]”. This should be rewritten.

The authors wrote “Our current tree (Figure 2) is congruent with Bond et al. (2014) in placing Paleocribellatae (Table 1, Hypochilus; Figure 1, Node 9) as sister to Neocribellatae.“ Actually, this is the result showed in Coddington' supertree (2005). Quite different, the authors results show, as Bond et al. (2014: fig. 2), a novel cluster formed by Hypochilus and Kukulcania. However, the relations between Hypochillidae, Austrochiloidea and Araneoclada cannot be solve in the absent of one of those representatives, in this manuscript, the lacking of any species of Austrochiloidea. Thus, here different phylogenetic scenarios may be possible. The authors may check, e.g., the alternative phylogenetic hypotheses in Lopardo et al. 2004 (figs. 19-21). Briefly, 1) Austrochiloidea could be polyphiletic (fig. 20) with Austrochilidae as sister of Entelegynae, and Gradungulidae as sister of the classical Haplogynae plus (Austrochilidae, Entelegynae); 2) or Austrochiloidea could be monophyletic (fig. 21) as sister to Entelegynae plus Synspermiata. Also, there is a recent phylogenetic analysis (Agnarsson et al. 2015: fig. 11) using 10 molecular markers that favors the polyphyly of Austrochiloidea. Thus, the authors must consider these other alternative hypotheses and discuss them, otherwise, they will be biased toward only one possible alternative as a consequence of low taxon sampling (in this case representatives of Austrochiloidea). Here is the reference to that paper:
Lopardo L, Ramirez MJ, Grismado C, Compagnucci LA. (2004) Web building behavior and the phylogeny of austrochiline spiders. Journal of Arachnology 32: 42–54.10.1636/H02-45

Theridiidae (line 481)
In Dimitrov et al. 2012 (fig. 1), Theridiidae also emerges as the sister group of the remaining araneoids with a 76 bootstrap support (highly supported). The authors should check this and rephrase the sentence.

Deinopoidea (line 494-496)
“[…] corroborates the non-monophyly of the classically defined Orbiculariae […]” This was also showed by Fernández et al. 2014 (fig. 2A, B). The authors should add this information.

“Deinopoidea, with these data, is polyphyletic.” This was also showed by Dimitrov et al. 2012 (fig. 1) and, as in this manuscript, the polyphyly of Deinopoidea was low supported. The authors should add this information.

“Instead, a new clade, Uloboridae + Oecobiidae, is sister to Araneoidea + Deinopidae + the RTA clade.” This affirmation is incorrect. The clade formed by Uloboridae + Oecobiidae (for clarity I call it here UO) is sister to the clade formed by Deinopoidea + the RTA clade (say DR). Both clades UO + DR forms a new clade, UO-DR, that is sister to Araneoidea + Eresidae (meaning Stegodyphus, call it AE). The authors should check this and rephrase the sentence.

Eresoidea+ Oecobiioidea (line 498)
The Eresoidea/Oecobiioidea issue was also discuss in previous phylogenetic studies (please see below for references). The authors’ results are consistent with those of Miller et al. (2010), and must be considered for the discussion. Note that Miller et al.' results were highly supported (Fig. 3).
Miller et al. (2010): "A series of phylogenetic studies suggested that Oecobiidae, Hersiliidae, and Eresidae form a clade, the Eresoidea (Agnarsson et al., 2006; Coddington et al., 2004; Coddington and Levi, 1991; Griswold et al., 1999, 2005; Platnick et al., 1991) based mostly on characteristics of the spinneret spigot morphology." [...] "All Bayesian and some parsimony analyses (Figs. 3, S1, S3–S5) placed Eresidae sister to nicodamids plus (at least araneoid) orb-weavers." [...] "Oecobiidae and Hersiliidae together form a clade, the Oecobiioidea." [...] "All Bayesian analyses place Oecobiioidea sister to the RTA clade (Figs. 3, S1, S3, S5); a close relationship between Eresidae and the Oecobiioidea is never supported." [...] "Some other investigators (e.g., Wunderlich, 2004) have not accepted a close relationship between Eresidae and Oecobiioidea."

RTA and Dionycha relationships (line 519)
“[…] traditional view of RTA phylogeny […]” The authors must add some references here. Are they referring as the traditional view of RTA phylogeny to Coddington and Levi 1991 (supertree), Coddington 2005 (supertree) or Griswold et al. 2005 (morphological phylogeny)? In those works, the internal relations of the RTA clade are not fully resolve.
Miller et al. 2010, however, provide a more comprehensive work to compare the RTA relationships as it present better taxon sampling and resolution.

Supertree considerations (line 523)
Coddington's (2005) figure 2.2 is a supertree (a summary of different phylogenetic analyses). In this case, as the authors are discussing and comparing data, I extremely recommend them to use source results as in Griswold et al. (2005). In this sense, readers may check and compare clades supports, tree resolution, phylogenetic analyses and characters scoring.

Amaurobiidae (line 527)
“[…] these data indicate a novel placement for Amaurobiidae.” This sentence is contradictory with the following sentence: "Our amaurobiid placement, basal to an agelenoid and dictynoid grouping corroborates previous findings (Miller et al., 2010; Spagna et al., 2010)." The authors should check this and rephrase the sentence.

Homalonychidae (line 535)
“[…]recover Homalonychidae (representing Zodarioidea).” This was corroborated by Miller et al. (2010): "Previous investigators have suggested that Chummidae and Homalonychidae might be close relatives of zodariids (Jocqué, 2001; Jocqué and Dippenaar-Schoeman, 2006; Roth, 1984). Homalonychids were found to be close to zodariids in some analyses. Chummids were never close to zodariids. Zodariids, penestomids, chummids and homalonychids have never been included in a quantitative higher-level phylogenetic analysis." It should be mention in the text.
Agnarsson et al. 2013 figure 11 show Zodarioidea (including Homalonychidae) as sister of Agroeca (Liocraniidae), and this genus sister to lycosoids and dionychans. In figure 12, however, Homalonychidae emerges as sister to lycosoids and dionychans. This should be mention by the authors. Also, is important to remark that, in both trees, the supports of those relations are low (less than 0.45 bootstrap proportion).

Conclusion (line 550)
“Following Coddington and Levi (1991), higher-level spider systematics underwent a series of challenges from quantitative studies of morphology, producing provocative but weakly-supported hypotheses. (Griswold et al., 1998, 2005).” I found this sentence very disturbing.
The higher-level spider phylogeny showed in Coddington and Levi (1991: fig. 2) is a supertree build by hand using different spider's phylogenies, which lacks any kind of statistical support (meaning sensitivity). Furthermore, none a single phylogenetic tree used to build that supertree have statistical supports. So, I'm wondering which the author’s concept of support is. Citing Grant and Kluge 2003 (Cladistics 19 379–418): "In systematics, sensitivity analysis is generally interpreted as providing a measure of support, where results that are insensitive (robust, stable) are considered well supported. Support, in turn, is almost universally taken to mean certainty, confidence, probability, or reliability [references provided in the paper]. [...] Or the concept of support can be salvaged generally as an indicator of evidential ambiguity and a report on the decisiveness of tests, where support is defined objectively as the degree to which critical evidence refutes competing hypotheses." In any case, Coddington and Levi (1991) supertree is not supported statistically and should be used as a framework (potential idea) of spiders’ relationship. When new methodologies and technologies appear, the knowledgeable (and probably new) evidence can be tested once again. In Griswold et al. 2005, for example, they tested the homology of several characters across the spider tree of life (impossible in a supertree) and explored/showed several clade support values (Bremer support, GC frequences and weighting character sensitivity). As the authors said, these examples differ in the topology of the tree. But, even if in the recent “quantitative studies of morphology (e.g. Griswold et al. 2005) present weakly supported hypotheses”, those hypotheses should be preferable to the old ones that lack any support at all.
The authors should check this and rephrase the sentence.

Additional comments

The manuscript presents a great amount of genomic data across the spiders’ tree of life. This novel knowledge is priceless and contributes greatly to spiders’ systematics and phylogenomics in general. This manuscript also presents several methodologies to treat the phylogenomic data that is more than desirable. However, as I described in the Validity of the Findings the manuscript presents several important problems in how the authors showed their results regarding previous spider’ phylogenetic studies, how they cited references, as well as how they discussed the spider’s tree of life. For those reasons, I will require a significant revision which I feel is major enough that I would need to re-evaluate any revised version. The authors may found several minor comments and suggestions on the attached file.

Reviewer 2 ·

Basic reporting

No comments.

Experimental design

The authors use 70 target taxa in their analysis, representing 52 spider families. This is about 45% of the total family diversity within the order Araneae. The design is ambitious and the authors have garnered an impressive amount of data. Nevertheless, the study, although a dramatic improvement over past studies investigating the phylogenetic relationships among spider families, still only includes less than half of the total family diversity.

The authors criticize past studies because in these other studies, “only a small percentage of spider families were sampled and monophyly of individual families could not be tested.” But their paper still only includes less than half the total family diversity and is also not testing monophyly of individual families since, for the most part, only a very few target taxa per family are included. I suggest the authors downplay this particular criticism of past studies. Their approach is most definitely a huge improvement, but more still needs to be done to capture the overall diversity present in this order.

It is also not completely clear how the authors chose the particular target species representing the “major lineages.” Why, for example, did they choose Cicurina to represent dictinids/dictynoids? It would be helpful for the authors to include more information on how and why individual target taxa representing the various lineages were chosen.

Their use of the newly constructed spider-specific core ortholog group is outstanding and will present an important and significant step forward for all future high-level phylogenomic studies in this group of organisms.

Validity of the findings

The authors say, “it is increasingly clear that taxon sampling and data quality are more important than quantity.” Yet when their intention is to explore the phylogenomics of an entire order of organisms (Araneae), I would argue that quantity, in the form of broad representation of the entire order (of all the families and all the lineages) is still important in studies such as this. Quantity should not be trivialized when the object is to understand the phylogeny of an order.

When addressing mygalomorph relationships in the discussion, the authors state, “While our sampling here and previously…is far greater than any other published phylogenomic study [on mygalomorphs]…taxon sampling remains insufficient to address major issues aside from deeper level phylogenetic problems.” I would suggest that this statement generally holds true for the entire study and should be pointed out in the summary.

Additional comments

This is an excellent and important contribution for the field of arachnology. It presents new techniques for addressing phylogenetic relationships within the order Araneae. My comments in all other sections of this review should be shared with the authors. My criticisms are minor and can be easily addressed by the addition of a table or a sentence or two explaining the choice of target taxa for the various lineages and by stressing that, although they have improved overall lineage representation, ideally, more target taxa should be included in future studies to adequately represent the entire diversity present in this order.

---

## Round 0.2 · accepted · Accept

· Academic Editor

Accept

Thank you for your thorough revisions. Also, thank you for the acknowledgement for comments and to the reviewers, anonymous or otherwise. I look forward to seeing your paper in "print".